# Learning from the electronic structure of molecules across the periodic table

**Manasa Kaniselvan** [1,2,†], **Benjamin Kurt Miller** [1], **Meng Gao** [1], **Juno Nam**[1,3,†] **& Daniel S. Levine**[1]

[1]FAIR at Meta    [3]MIT    [2]ETH Zurich
[†]Work done during internship at FAIR
mkaniselvan@ethz.ch, levineds@meta.com

## Abstract

Machine-Learned Interatomic Potentials (MLIPs) require vast amounts of atomic structure data to learn forces and energies, and their performance continues to improve with training set size. Meanwhile, the even greater quantities of accompanying data in the Hamiltonian matrix $\mathbf{H}$ behind these datasets has so far gone unused for this purpose. Here, we provide a recipe for integrating the orbital interaction data within $\mathbf{H}$ towards training pipelines for atomic-level properties. We first introduce HELM ("Hamiltonian-trained Electronic-structure Learning for Molecules"), a state-of-the-art Hamiltonian prediction model which bridges the gap between Hamiltonian prediction and universal MLIPs by scaling to $\mathbf{H}$ of structures with 100+ atoms, high elemental diversity, and large basis sets including diffuse functions. To accompany HELM, we release a curated Hamiltonian matrix dataset, 'OMol_CSH_58k', with unprecedented elemental diversity (58 elements), molecular size (up to 150 atoms), and basis set (def2-TZVPD). Finally, we introduce 'Hamiltonian pretraining' as a method to extract meaningful descriptors of atomic environments even from a limited number atomic structures, and repurpose this shared embedding space to improve performance on energy-prediction in low-data regimes. Our results highlight the use of electronic interactions as a rich and transferable data source for representing chemical space.

## 1 Introduction

The availability of large volumes of high quality training data has been key to advancing the state of structure-property prediction models, particularly for applications as Machine-Learned Interatomic Potentials (MLIPs). While MLIPs trained on sizable datasets can now learn the mappings between atomic positions and inter-atomic forces and energies in a variety of structures, challenges remain when (1) generalizing to highly out-of-distribution structures (Antoniuk et al., 2025; Sheriff et al., 2025; Zhai et al., 2023) and complex materials (Liu et al., 2024; Miret et al., 2025), (2) satisfying the physics required to compute accurate vibrational properties (Fu et al., 2025) and realistic Molecular Dynamics (MD) trajectories (Fu et al., 2022; Bigi et al., 2024), and (3) working with the limited quantity of training data available at more chemically accurate levels of theory. Despite the advent of very large public datasets of DFT calculations, the performance of state-of-the-art MLIPs is still highly data-limited (Wood et al., 2025), and is expected to improve as more data becomes available. However, as the current largest models are already trained on over 10 billion core hours worth of DFT data (Levine et al., 2025), further increases in dataset size may not be practically feasible. Alternative sources of data are thus required to further advance MLIP development.

Generating this (typically Density Functional Theory (DFT)-level) training data requires computing the electronic Hamiltonian matrix for each given system of atoms. *For every system of $N$ atoms, there is 1 energy label, $\mathcal{O}(N)$ force labels, and $\mathcal{O}(N^2)$ labels in the Hamiltonian matrix.* The latter has so far not been leveraged towards training large-scale atomic property prediction models, although they are available from DFT calculations with no additional compute.

At the same time, the Hamiltonian matrix contains far more information than the atomic-level properties (e.g., forces, energies) which are computed from it. It includes information about excited states, ionization energies, electron density, multipole moments, and more. Incorporating this elec-

tronic structure data in different ways has served to improve data efficiency and generalizability across atomic-level prediction tasks and challenging interaction regimes, by exploiting patterns in the underlying electronic representations of atomic interactions (del Rio et al., 2023; Foster et al., 2025; Kang et al., 2025; Suman et al., 2025; Sunshine et al., 2023; Tang et al., 2025; Shao et al., 2023). These initial attempts to inform MLIPs with electronic interactions have, however, been limited to demonstrations on relatively restricted datasets in terms of both molecular size and elemental composition. Existing models struggle to scale to the structure sizes (Xia et al., 2025), basis sets (Li et al., 2025), and number of atomic elements necessary to advance the state of the art of MLIPs. Here we directly address these challenges to enable scalable electronic-structure learning, and provide a recipe for learning energies from a shared embedding space via 'Hamiltonian pretraining'. We make the following contributions:

1. Develop HELM - "Hamiltonian-trained Electronic-structure Learning for Molecules", a scalable Hamiltonian matrix prediction network which enables transfer-learning to energy prediction tasks from a unified atomic orbital basis. HELM is state-of-the-art on the most challenging Hamiltonian matrix benchmarks openly available, and is the first 'universal' electronic structure prediction model demonstrated on structure sizes and chemical diversity comparable to those treated by current MLIPs.

2. Introduce a curated dataset of Hamiltonian matrices, 'OMol_CSH_58k' based on the Open Molecules (OMol25) (Levine et al., 2025) data. Compared to the previous largest publicly-available Hamiltonian matrix benchmark (Khrabrov et al., 2024), OMol_CSH_58k efficiently samples electronic interactions across an unprecedented range of atomic elements, inter-atomic distances, and orbital types, matching the scale and diversity of current atomic force and energy datasets. It presents a realistic and challenging target for Hamiltonian pretraining approaches.

Finally, we explore the use of HELM towards improving energy prediction via 'Hamiltonian pretraining', using OMol_CSH_58k as well as the second-largest publicly available Hamiltonian database, $\nabla^2$DFT (Khrabrov et al., 2024). We find that embeddings after Hamiltonian matrix training contain fine-grained representations of atomic environments, while this structure does not appear when training directly on energy labels for the same number of molecules. Hamiltonian pretraining thus leverages these representations to achieve up to a $2\times$ improvement in energy-prediction accuracy in low-data controlled experiments. Achieving such improvements by naively scaling atomic force and energy data alone would require more than an order of magnitude more calculations.

## 2 BACKGROUND AND PREVIOUS WORK

Many of the properties being investigated in computational chemistry research are computed using first-principles electronic structure methods such as DFT, which iteratively finds a consistent representation of electron density and potential for a system of atoms, defined by atomic elements and positions (Kohn & Sham, 1965). The computed electronic structure information can then be processed to directly obtain various quantities at the electronic and atomic levels, such as molecular energies (see Section A.2).

**Learning interatomic potentials:** MLIPs approximate the electronic Born–Oppenheimer potential energy surface at far lower cost, enabling large-scale MD and high-throughput virtual screening of molecules and materials. Early work moved from empirical force fields with hand-crafted terms to data-driven models that learn energies and forces from DFT, using local descriptors with neural networks or Gaussian processes. To improve transferability and data efficiency, descriptors were designed to encode permutation invariance and rotational equivariance (Behler & Parrinello, 2007; Bartók et al., 2010; Drautz, 2019).

Motivated by orbital symmetries in DFT, modern MLIPs adopt graph neural networks that maintain rotational equivariance by constructing features in the form of irreducible representations ('irreps'), and mixing them through message passing iterations performed with tensor products (Anderson et al., 2019; Batzner et al., 2022; Batatia et al., 2022). In these networks, 'descriptors' are effectively learned through data. On this foundation, universal MLIPs pretrain on large, heterogeneous datasets to yield single models usable across diverse elements over the periodic table (Chen & Ong, 2022; Deng et al., 2023; Batatia et al., 2023; Park et al., 2024; Yang et al., 2024; Rhodes et al., 2025). Recently, Meta's UMA model (Wood et al., 2025) demonstrated multi-task training across molecules

and materials to span broad chemical space and achieve strong accuracy and efficiency. However, despite being trained on over 459M energy labels, UMA's performance is still limited by data.

To close the remaining gap to first-principles fidelity, incorporating electronic structure signals may be an essential next step, both in terms of the additional volume of data available at electronic resolution, and the rich information contained within it. Long-range electrostatics, dispersion, non-local charge transfer, and field response remain challenging for current models, and initial explorations that learn auxiliary electronic observables or responses provide encouraging signals (Anstine et al., 2025; Falletta et al., 2025; Cheng, 2025).

**Learning orbital interactions**: Symmetry-constrained GNN architectures have recently been adapted to instead directly predict the underlying electronic structure of atomic systems. Here, the most common objective is to learn the elements of the ground state Hamiltonian matrix $\mathbf{H}$ (also referred to as the "Fock" matrix in some contexts). It has conventionally been used to study the band structure (Fischetti & Laux, 1996) and non-equilibrium transport properties of materials (Brandbyge et al., 2002), or serve as input to quantum many-body solvers (Kolorenč et al., 2010). In such electronic structure prediction models, each sub-matrix of the Hamiltonian, which represents interactions between atomic orbitals of angular degree $l_1$ and $l_2$, is decomposed and transformed into a direct sum of irreps of degree $|l_1 - l_2|$ to $l_1 + l_2$: $(l_1 \otimes l_2 = |l_1 - l_2| \oplus (|l_1 - l_2| + 1) \oplus \cdots \oplus (l_1 + l_2))$ using the Clebsch–Gordan (CG) coefficients. As many equivariant GNNs already operate in a basis of spherical harmonic coefficients (typically mapping the output $l = 0$ or $l = 1$ coefficients of output embeddings to atomic energies and forces (Fu et al., 2025)) they can be adapted to instead map the full set of $l = 0...l_{max}$ coefficients to every submatrix of $\mathbf{H}$.

To learn the $\mathbf{H}$ of small molecules, initial models used full tensor products to mix spherical harmonic coefficients while maintaining rotational equivariance (Unke et al., 2021; Gong et al., 2023). Following work focused on eliminating unnecessary tensor product components and taking advantage of CG coefficient sparsity to improve the efficiency of these operations (Yu et al., 2023b; Gong et al., 2023; Luo et al., 2025). More recent models have replaced full tensor products by SO(2) convolution approaches (Passaro & Zitnick, 2023) to decrease the computational complexity of treating high $l_{max}$ from $\mathcal{O}(l_{\max}^6)$ to $\mathcal{O}(l_{\max}^3)$ (Li et al., 2025; Yu et al., 2025) - this is critical for treating interactions within basis sets containing $d$ and $f$ orbitals. Several works are now integrating electronic structure prediction models with custom datasets to enable new research directions in electronic materials which were previously computationally unfeasible (Xia et al., 2025; Haldar et al., 2025).

## 3 METHODS

We introduce HELM: "Hamiltonian-trained Electronic-structure Learning for Molecules." HELM is composed of a feature-extraction backbone with shared weights and two separate output heads

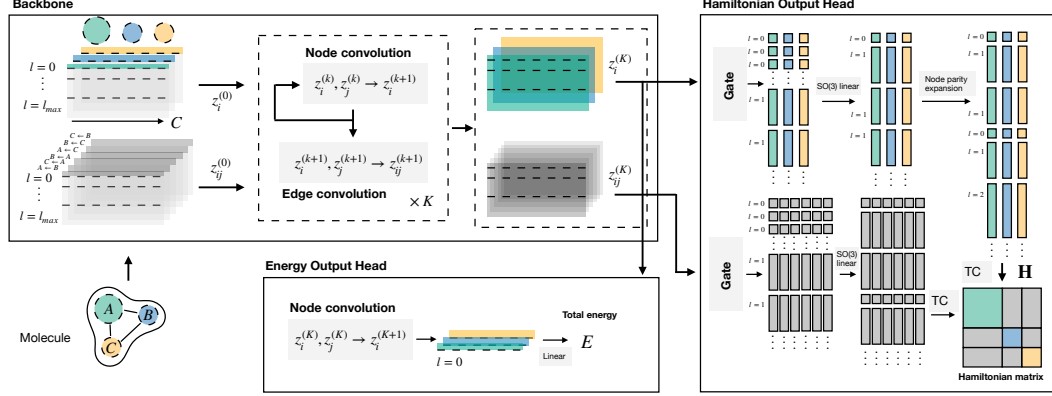

Figure 1: Architecture of HELM. Each rectangle in the backbone represents an embedding for an atom or inter-atomic interaction, has a shape of $(l_{\max} + 1)^2 \times C$, and contains a set of learnable spherical harmonic coefficients for each $(l, m)$ up to a specified $l_{max}$. TC = Tensor construction layer to reconstruct the Hamiltonian (see **Section B.1**).

(Fig. 1). One head is designed to predict the Hamiltonian matrix $\mathbf{H}$, while the other predicts total energy values from a given atomic structure. In a typical training regime, HELM is first trained to estimate $\mathbf{H}$, then the backbone features are used directly and/or finetuned to estimate energy.

**Backbone:** The backbone of HELM is an equivariant message passing GNN based on that of Fu et al. (2025). The input to HELM is a molecular graph, represented by displacement vectors $\boldsymbol{r}_{ij} \in \mathbb{R}^3$ and a one-hot array of atom types $Z_i \in \{0, 1\}^h$ with $i, j = 1, \ldots, N$ where $N$ is the number of atoms, and $h$ is the maximum atomic number. Unlike MLIPs, where predictions are made nodewise, learning Hamiltonian matrices requires the model to make predictions for both nodes (intra-atomic orbital interactions) and directed edges (inter-atomic orbital interactions). We initialize node and edge embeddings $z_{ij}$ as follows:

$$z_{ij}^{(1)} = f_{\text{init}}^{\theta_e, \theta_g}(Z_i, \boldsymbol{r}_{ij}) = \begin{cases} f_{\text{embed}}^{\theta_e}(Z_i) & \text{if } i = j \\ f_{\text{gauss}}^{\theta_g}(\boldsymbol{r}_{ij}) & \text{if } i \neq j, \|\boldsymbol{r}_{ij}\| < r_{\text{cut}} \end{cases}, \quad [z_i^{(k)}, z_{ij}^{(k)}] = \left(z_{ij}^{(k)}\right)_{i,j=1}^N \quad (1)$$

with $k = 1, \ldots, K$ where $[z_i^{(k)}, z_{ij}^{(k)}]$ denotes an $N \times N$ sparse array of multi-channel spherical harmonic coefficient embeddings, and $j$ indexes over atoms within $r_{cut}$ of $i$. Each of these embeddings has shape $(l_{\max} + 1)^2 \times C$, where $C$ is a channel hyperparameter. $f_{\text{embed}}^{\theta_e}$ and $f_{\text{gauss}}^{\theta_g}$ are atomic element and gaussian radial embeddings, and $l_{\max}$ is the spherical harmonic degree required to represent all coupled interactions in the orbital basis. These initial embeddings are transformed through $K$ layers: With $y_i^{(k)} = \sum_j f_{\text{node}}^{\theta_n^k}\left([z_i^{(k)}, z_j^{(k)}], \boldsymbol{r}_{ij}\right)$,

$$z_{ij}^{(k+1)} = f_{\text{layer}}^{\theta_n^k, \theta_d^k}\left([z_i^{(k)}, z_j^{(k)}], \boldsymbol{r}_{ij}\right) = \begin{cases} y_i^{(k)} & \text{if } i = j \\ f_{\text{edge}}^{\theta_d^k}\left([y_i^{(k)}, y_j^{(k)}], \boldsymbol{r}_{ij}\right) & \text{if } i \neq j, (\|\boldsymbol{r}_{ij}\| < r_{\text{cut}}) \end{cases}, \quad (2)$$

where $f_{\text{node}}^{\theta_n^k}$ and $f_{\text{edge}}^{\theta_d^k}$ are embedding update operations, internally consisting of SO(2) convolutions and gating mechanisms (Passaro & Zitnick, 2023). $\theta_e, \theta_g, \theta_n^1, \theta_d^1, \ldots, \theta_n^k, \theta_d^k$ are learnable weights. The key idea is that the node embeddings at layer $k + 1$ are a function of node embeddings at layer $k$; however, edge embeddings at layer $k$ are a function of node embeddings in the same layer $k$, specifically the nodes they connect. This one-way dependence of edge embeddings on node embeddings serves two purposes. First, during training, it incorporates information about the $1/r$ decay of Coulombic electron integrals present in the interatomic interactions of $\mathbf{H}$ (see Section B.2). Second, it allows us to omit the edge updates when re-purposing the trained backbone for energy prediction. We denote the full backbone, and also just the diagonal, with ($\circ$ is function composition):

$$f_{[z_i, z_{ij}]}^{\theta} = f_{\text{layer}} \circ \overset{K-2}{\cdots} \circ f_{\text{layer}} \circ f_{\text{init}}, \quad f_{\text{diagonal}}(x) = (x)_{i=j=1}^N, \quad f_{[z_i]}^{\theta} = f_{\text{diagonal}} \circ f_{[z_i, z_{ij}]}^{\theta}, \quad (3)$$

**Hamiltonian output head** ($\mathbf{H} = f_{\mathbf{H}}^{\theta}([z_i^{(K)}, z_{ij}^{(K)}])$)**:** Each block of $\mathbf{H}$ can be decomposed into a direct sum of irreps (see Section B.1), which serve as the labels for every node and edge of the atomic graph. The Hamiltonian output head then maps each of the learned embeddings $z_i^{(K)}$ and $z_{ij}^{(K)}$ to the corresponding label extracted from $\mathbf{H}$. We make two additional architectural choices designed to improve HELM's ability to handle larger Hamiltonian datasets:

(1) To better distinguish different orbital shells when using large basis sets (e.g. multiple $d$-type functions for a given principal quantum number), we add additional gated nonlinearity (Weiler et al., 2018). Each irrep of nonzero degree is multiplied by a learned scalar, $g_k$ passed through a sigmoid activation $\sigma$: $(\bigoplus \sigma(z_{ij,l=0}^{(K)})) \oplus (\bigoplus \sigma(g_k) z_{ij,l \neq 0}^{(K)})$.

(2) The irreps in the diagonal blocks of $\mathbf{H}$, which represent interactions between orbitals on the same atom have additional parity constraints: $\ell_1^A \otimes \ell_2^A = \bigoplus_{\ell_3^A = |\ell_1^A - \ell_2^A|}^{\ell_1^A + \ell_2^A} \left[ \delta_{(-1)^{\ell_1^A + \ell_2^A}, (-1)^{\ell_3^A}} \cdot \ell_3^A \right]$. Here, $\ell_1^A, \ell_2^A$ represent the degree of the interacting orbitals within atom $A$, $\delta$ is the Kronecker delta, and the $\ell_3^A$s represent their tensor product in the coupled basis. To take advantage of constraints, we compute only unique and non-zero node values.

**Energy output head** ($E = f_E^\theta(z_i^{(K)})$): While the total energy $E$ of a molecule can be directly computed from $\mathbf{H}$[1], this process involves re-assembling the predicted $\mathbf{H}$ from the internal representation used in HELM. This makes the use of gradient-based force prediction or direct energy finetuning computationally expensive for large structures. Additionally, some of the additional DFT quantities (namely the evaluation of the exchange-correlation energy) require numerical evaluation of the density obtained from $\mathbf{H}$ on a grid and is thus considerably slower than a forward-pass of the network. We therefore design an additional output head to learn a mapping $E = f_E^\theta([z_i^{(K)}])$ directly from the final node embeddings. In practice, $f_E^\theta([z_i^{(K)}])$ is implemented with a single embedding update block, followed by a linear transformation $\mathbf{w}$ of the $l = 0$ (scalar) output components, and a sum over the nodes in each molecule to obtain a total energy prediction $E = \sum_{i=1}^{N} \mathbf{w}^\top z_{i,\,l=0}^{(K+1)}$.

## 3.1 OMOL CLOSED-SHELL 58K

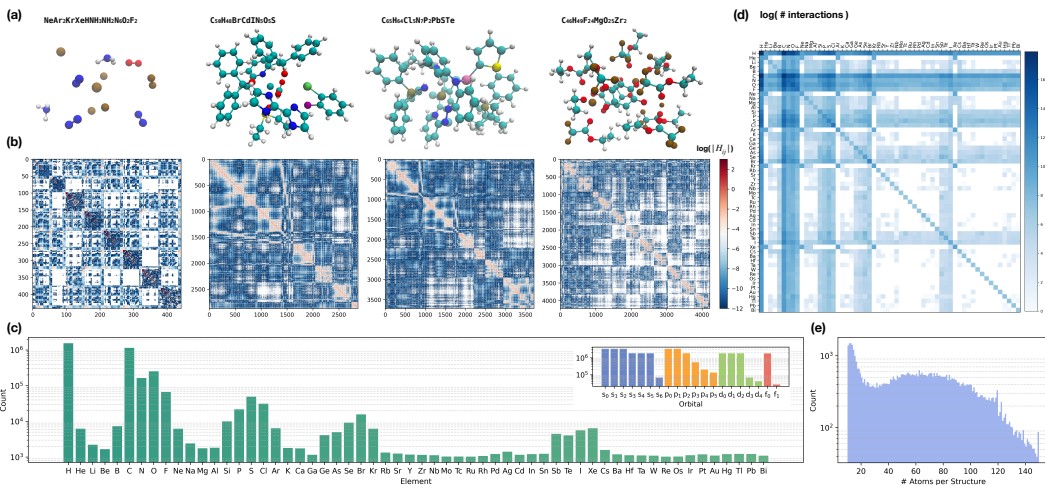

Figure 2: Breakdown of atomic and electronic interactions within the OMol_CSH_58k dataset. (a) Four sample molecules of increasing size and (b) magnitude plots of their Hamiltonian matrices. (c) The full dataset spans 58 atomic elements, as shown in the element distribution histogram; the inset details the corresponding orbital count within the def2-TZVPD basis across the dataset. (d) Element—element interaction heatmap indicating the diversity of interactions across the dataset. (e) Histogram of the distribution of atoms per structure (ranging from 10 to 150).

Although Hamiltonian datasets for molecules have become increasingly diverse (Schütt et al., 2019a; Yu et al., 2023a; Khrabrov et al., 2024), they remain heavily limited in chemical scope compared to recent datasets of atomic forces and energies (Levine et al., 2025). Achieving 'universal' electronic structure learning and potential transferability to other tasks via pretraining requires data with sufficient sampling of both electronic structures and atomic energies for different atomic elements, structure sizes, configurations, and interaction distances.

To bridge this gap in chemical diversity and complexity between electronic and atomic property datasets for molecules, we introduce 'OMol_CSH_58k', a curated subset of the more than 110M entries in the Open Molecules 2025 (OMol25) dataset which contains both electronic structure (KS Hamiltonian) data as well as atomic forces and energies for over 110M structures. This subset contains 56,657 entries. OMol_CSH_58k is designed to efficiently sample three degrees of freedom which generate maximally informative training structures while avoiding redundancy in orbital interactions: (1) atomic elements, (2) orbital interactions, and (3) inter-atomic distance. The dataset includes 58 elements, which comprise all the first 83 atomic elements other than the first-row transition metals and the lanthanides.[2] It has roughly 1000 occurrences of all heavy elements. Each

---

[1]Given the availability of a few other non-iteratively-computed DFT quantities (see Section A.2)

[2]These two sets of elements were excluded because they contain very high angular momentum orbitals ($g$-functions) in the OMol basis set (def2-TZVPD) which, while posing no technical challenge to the model framework, do dramatically increase memory consumption during training.

structure contains between 10 and 150 atoms, and interaction distances within the largest structures range up to 15 Å.[3] With $> 50k$ structures, 58 atomic elements, and long-range interactions, OMol_CSH_58k is the largest Hamiltonian matrix dataset to date in structure size, matrix size, and number of elements.

The presence of heavy atoms and large structures make this dataset especially challenging for Hamiltonian matrix prediction, by requiring the treatment of higher $l_{max}$ (6 to consider interactions between $f$-orbitals), high orbital multiplicity, and large target sizes to account for the multiple compact and diffuse orbitals in the basis. We introduce two test splits to assess the performance of HELM and future models: "OMol_all_5k" and "OMol_common_1k". The first consists of 4,937 structures that are drawn from the OMol25 test data, and therefore have unique compositions from all training data. These data span the same elemental composition as the OMol_CSH_58k dataset. The second split consists of 1,006 structures also drawn from the OMol25 test data, but only containing the elements present in $\nabla^2$DFT (H, C, N, O, F, S, Cl, Br). This allows for easier comparison of performance to existing models and datasets. An overview of the molecules and corresponding Hamiltonian matrices contained within the train and test splits is summarized in Table 1.

| | Elements | # Molecules | $\overline{N}_{\text{atoms}}$ | $\overline{N}_{\text{edges}}$ | # $H_{ii}$ | # $H_{ij}$ | E |
|---|---|---|---|---|---|---|---|
| OMol_CSH_58k | all 58 | 56,657 | 59 | 4,258 | 3,362,755 | 241,227,574 | ✓ |
| OMol_common_1k | H, C, N, O, F, S, Cl, Br | 1,006 | 98 | 9,086 | 98,399 | 9,140,166 | ✓ |
| OMol_all_5k | all 58 | 4,937 | 75 | 7,396 | 370,924 | 36,514,364 | ✓ |

Table 1: Summary of the OMol_CSH_58k train and test datasets, including the number of molecules, average number of atoms ($\overline{N}_{\text{atoms}}$) and edges ($\overline{N}_{\text{edges}}$) per molecule, and the total number of intra-atomic ($H_{ii}$) and inter-atomic ($H_{ij}$) orbital interaction matrices over all molecules in the datasets. The last column indicates the presence of total energy values per molecule.

**Loss and scalar referencing**: Previous Hamiltonian prediction networks treating small molecules have typically computed the loss with combinations of mean absolute errors (MAE) or mean squared errors (MSE) over the matrix elements (Yu et al., 2023b; Luo et al., 2025). However, straightforward element-wise losses are not comparable across different structure sizes and atomic elements, making it difficult to develop a meaningful training signal for large datasets with both size and element diversity. To address this, Li et al. (2025) introduced the 'wavefunction alignment loss', which effectively re-weights matrix elements according to their relative importance to the eigenvalue spectra. However, implementing the wavefunction alignment loss in practice involves re-assembling $\mathbf{H}$ after every forward pass during training. When treating large structures with 100+ atoms, building these intermediate matrices becomes a computational bottleneck.

Here, we posit an additional two reasons why naive element-wise losses may fail. First, as the number of heavy elements increases, the distribution of magnitudes in the matrix elements of $\mathbf{H}$ increases disproportionately. The parallel problem in energy prediction has typically been mitigated by computing reference values per atomic element, and using them to rescale the training data. Second, MAE losses are performed component-wise, which may bias the loss computed over otherwise rotationally-identical edges (Wang et al., 2024). To mitigate these effects, we first pre-process the node labels to scale and center the $l = 0$ components, which stem roughly from element-specific localized core electron states (see Section B.3). This follows a similar process applied in other recent work on Hamiltonian prediction for materials (Haldar et al., 2025; Qian et al., 2025), and serves to effectively homogenize the variance of the data across different elements. Finally, we train with a root-MSE+MSE loss over the labels, which is independent of irrep orientation.

## 4 RESULTS

In Section 4.1, we evaluate the ability of HELM to learn Hamiltonian matrix elements for two existing benchmarks, and show that it is state-of-the-art for this task. In Section 4.2, we use the backbone of HELM, pretrained on the Hamiltonian matrices, to perform low-data controlled training experiments on energy labels in the $\nabla^2$DFT and OMol_CSH_58k datasets, demonstrating an increase in per-molecule data efficiency stemming from pretraining.

---

[3]In practice, we clip the matrices to a maximum inter-atomic interaction distances of 12 Å, which is sufficient to capture matrix elements greater than $1 \times 10^{-5}$ (see Section B.2)

## 4.1 HAMILTONIAN MATRIX BENCHMARKS

**Table 2** summarizes results for benchmarks on two previous datasets, MD17/QM7 and $\nabla^2$DFT. Note that matrix element losses are not comparable across different datasets due to the difference in molecule size and elemental diversity. Prediction of the overlap matrix is omitted, as these are rapidly obtained with PySCF. Model/training parameters are in **Section C.1**/ **Section C.2**.

| | **MD17** | | | | **$\nabla^2$DFT** | | |
|---|---|---|---|---|---|---|---|
| | Water (3) | Ethanol (9) | Malondialdehyde (9) | Uracil (12) | Train/Test-2k (39/37) | Train/Test-5k (41/38) | Train/Test-10k (42/39) |
| | $H^{err}$ ($\times 10^{-6} E_h$) | | | | $H^{err}$ ($\times 10^{-6} E_h$) | | |
| SchNOrb | 165.4 | 187.4 | 191.1 | 227.8 | 21500.0 | 20700.0 | 20700.0 |
| PhiSNet | 17.59 | 12.15 | 12.32 | 10.73 | 180 | 330.0 | 350.0 |
| QHNet | 10.79 | 20.91 | 21.52 | 20.12 | 840.0 | 730.0 | 520.0 |
| SPHNet | 23.18 | 21.02 | 20.67 | 19.36 | – | – | – |
| HELM | **9.33** | **5.79** | **4.86** | **3.61** | **60.33** | **57.41** | **59.21** |
| | $E^{err}$ (meV) | | | | $E^{err}$ (meV) | | |
| HELM | 23.67 | 28.93 | 20.15 | 22.69 | 440.82 | 399.840 | 415.55 |

Table 2: Comparison of **H** matrix element prediction error on the MD17 and $\nabla^2$DFT molecule datasets, for SchNOrb (Schütt et al., 2019b), PhiSNet (Unke et al., 2021), QHNet (Yu et al., 2023b), and SPHNet (Luo et al., 2025). In parentheses under each dataset are the average number of atoms per molecule. Element-wise matrix errors are computed as $H^{err} = \frac{1}{N} \sum_{i,j} |H_{ij}^{\text{pred}} - H_{ij}^{\text{ref}}|$. The best performing model is indicated in bold, and the second best is underlined. Benchmark numbers for $\nabla^2$DFT are taken from Khrabrov et al. (2024). In the last row, we summarize the MAE energy error ($E^{err}$) computed from $H_{ij}^{\text{pred}}$ and $H_{ij}^{\text{ref}}$ using PySCF, with the functional specific to each dataset.

**MD17/QM7** is a commonly considered small-molecule benchmark for Hamiltonian prediction. It tests generalization to conformers of single molecules with the same set of atomic elements (Schütt et al., 2019b). The dataset consists of four small molecules (water, uracil, ethanol, malondialdehyde) evolving along a Molecular Dynamics trajectory, with Hamiltonian matrices in a def2-SVP basis provided for each snapshot. When training on the Hamiltonian matrices of these molecules, HELM is competitive with all previous 'direct' (single evaluation) models [4], achieving state-of-the-art accuracies in the MAE over matrix elements ($H^{err}$). When using the predicted **H** to recompute the total energy (see **Section A.2**), we can reproduce the total energy computed from the reference matrices to within 30 meV per molecule.

$\nabla^2$**DFT** is currently the largest openly available molecule benchmark with electronic structure information (Khrabrov et al., 2024). It contains drug-like molecules with C, N, S, O, F, Cl, Br, and H atom types, up to 62 atoms per structure, computed at the $\omega$B97X-D/def2-SVP DFT level of theory. Multiple low-energy conformations with atomic forces, energies, and Hamiltonian matrices are provided for each molecule. In **Table 2**, we compare performance on the 2k, 5k, and 10k train/test splits with those of previous models. HELM predicts matrix elements to $\sim 60\mu E_h$, which is $\sim$3-5$\times$ better than those of the previous best-performing model. Further analysis of the connection between matrix element errors, energy eigenvalues, and total energy errors are provided in **Section D.1**.

## 4.2 HAMILTONIAN PRETRAINING

We demonstrate the benefit of Hamiltonian pretraining with controlled experiments on a limited volume of energy data (e.g., the energy labels available for $\nabla^2$DFT's 2k, 5k, and 10k splits, and OMol_CSH_58k). $\nabla^2$DFT's differently-sized conformations train and test splits allow for an investigation of scaling with data, while OMol_CSH_58k presents a significantly more challenging learning task, as it contains a highly limited quantity of energy labels for considerably larger structures and increased elemental diversity.

---

[4]A recent work adapting the QHNet architecture with a flow-matching training process and eigenvalue-based loss function has extended this accuracy to below 5 $\mu E_h$ (Kim et al., 2025). This flow-matching approach can be applied on top of any Hamiltonian prediction method to improve its accuracy, albeit with increased cost. As such, we focus on the underlying architectures.

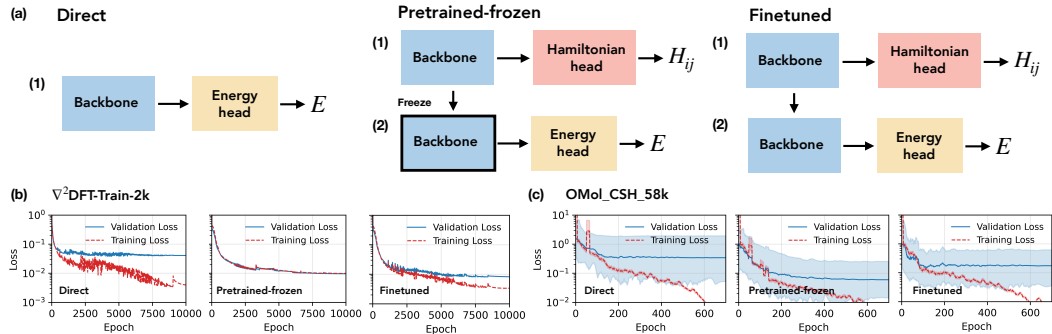

Figure 3: (a) The three training schemes we use to investigate the effect of Hamiltonian pretraining on the backbone of HELM for downstream energy prediction. (b) and (c) contain the corresponding loss over epochs for each of these schemes, when applied to $\nabla^2$DFT's train-2k split and OMol_CSH_58k. For the latter, the fill-between indicates a distribution of loss over each batch.

We compare three cases: first, we train the model (backbone and output head) on energies - this is the "direct" case. Second, we train HELM on Hamiltonian matrices, detach the backbone, freeze its weights, and connect it to a trainable energy output head which we subsequently train on scalar energy labels. We refer to this as the "pretrained-frozen" model. Third, we start from the same Hamiltonian-pretrained backbone model, connect to an energy output head, and finetune on energy data. The total number of parameters in the backbone and output head are equal in all three cases, and the number of learnable parameters are equal for the direct and finetuned models. Model sizes and training parameters are specified in Table 6 and Table 8. For a fair comparison, we used a fixed number of epochs and initial learning rate with a cosine scheduler for each dataset. Illustrations of the three training schemes, along with training loss plots, are shown in Fig. 3. Train/test dataset evaluation results are summarized in Table 3 and Table 4. We additionally report matrix element errors ($H^{err}$) achieved after the pretraining phase on OMol_CSH_58k for the two test splits (example matrix predictions are presented in Section D.2). The increased $H^{err}$ for OMol_all_5k primarily highlights the variation in matrix element magnitudes resulting from the presence of heavy elements.

| Split | | # Molecules | $E^{err}$ (meV) | | |
|---|---|---|---|---|---|
| | | | Direct | Pretrained-frozen | Finetuned |
| **2k** | Train | 8,097 | $97.16 \pm 4.12$ | $217.58 \pm 5.17$ | $76.30 \pm 1.56$ |
| | Test | 2,774 | $791.02 \pm 40.68$ | $324.64 \pm 27.41$ | $266.23 \pm 24.34$ |
| **5k** | Train | 18,908 | $240.631 \pm 6.20$ | $264.91 \pm 4.79$ | $116.52 \pm 1.35$ |
| | Test | 5,634 | $592.54 \pm 15.69$ | $303.92 \pm 7.92$ | $199.50 \pm 6.15$ |
| **10k** | Train | 33,150 | $285.39 \pm 4.23$ | $234.91 \pm 2.77$ | $176.75 \pm 1.37$ |
| | Test | 9,532 | $506.80 \pm 12.06$ | $265.91 \pm 7.86$ | $198.40 \pm 5.76$ |

Table 3: Energy performance metrics (MAE) for the direct, pretrained-frozen, and finetuned HELM models on the different conformers splits of $\nabla^2$DFT, along with 95% confidence intervals.

| HELM composition | Metric | Training method | OMol_CSH_58k | OMol_common_1k | OMol_all_5k |
|---|---|---|---|---|---|
| $\mathbf{H} = f_{\mathbf{H}}^{\theta} \circ f_{[\mathbf{z}_i, \mathbf{z}_{ij}]}^{\theta}$ | $H^{err}$ ($\mu E_h$) | - | | - | 961.64 | 2122.19 |
| $E = f_E^{\theta} \circ f_{[\mathbf{z}_i]}^{\theta}$ | $E^{err}$ (meV) | Direct | $353.15 \pm 15.17$ | $3631.69 \pm 227.36$ | $5976.40 \pm 209.33$ |
| | $E^{err}$ (meV) | Pretrained-frozen | $361.98 \pm 6.10$ | $2119.76 \pm 178.64$ | $3255.69 \pm 154.70$ |
| | $E^{err}$ (meV) | Finetuned | $254.44 \pm 9.68$ | $2533.22 \pm 172.14$ | $3926.12 \pm 155.21$ |

Table 4: Performance metrics for Energy for HELM trained on OMol_CSH_58k. Energy errors are reported with 95% confidence intervals.

In all cases, the "direct" model rapidly over-fits to limited number of energy labels in the training data. For $\nabla^2$DFT, the "pretrained-frozen" and "finetuned" models show both reduced overfitting and improved test accuracy, with the relative improvement in test accuracy from Hamiltonian pretraining

being highest for the smallest conformer split (Train/Test-2k). Note that despite using a limited quantity of energy data, the energy errors are already lower than those computed directly from the Hamiltonian matrices in **Table 2** - computing the energy directly from the Hamiltonian is highly sensitive to small perturbations in the matrix elements (see **Fig. 10**), and finetuning on high-precision energy labels helps bridge much of the remaining accuracy. The numerical improvements from pretraining are lower for the OMol test splits, as the pretrained-frozen and finetuned models also exhibit some degree of overfitting to the training data (**Fig. 3**). This likely stems from the increased diversity of atomic elements, and thus reduced sampling of energy labels over heavier elements. In this case, the pretrained-frozen model shows a $\sim 1.8\times$ improvement over the direct model. Finetuning increases the extent of overfitting, and shows a slightly lower improvement of $\sim 1.5\times$.

To visualize the effects of Hamiltonian pretraining, we extract and analyze the backbone embeddings from all three models when processing the first 250 molecules in OMol_CSH_58k. In **Fig. 4** we plot the UMAP analysis over the norm of each individual irrep in this set of embeddings (McInnes et al., 2018), colored by atomic element. The (1) dramatic increase in the number of distinct clusters within the "pretrained-frozen" embeddings as opposed to those of the "direct" model, as well as the (2) clearer distinction of heavier atomic elements which are less well-represented in the datasets (as opposed to Hydrogen ○ and Carbon ◉), suggest that more detailed descriptors of atomic environments are being learned from the Hamiltonian matrix data. Finetuning on OMol_CSH_58k visibly diminishes this fine-grained structure, while finetuning on $\nabla^2$DFT energy data preserves it (an extended analysis is in **Section D.4**), consistent with their relative trends in test accuracy.

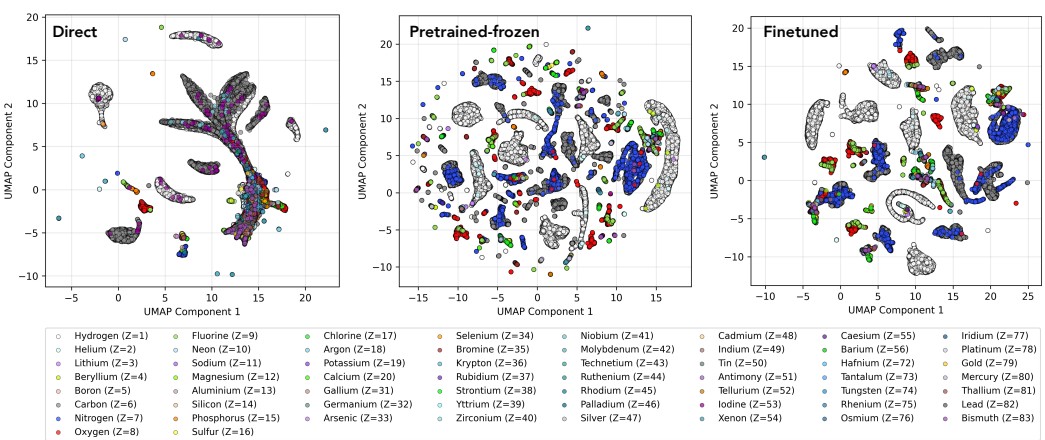

Figure 4: UMAP visualization of the output node embeddings ($z_i^{(K)}$) of the first 250 molecules of OMol_CSH_58k, extracted from the direct, pretrained-frozen, or finetuned models, and colored by element. There are 100,233 datapoints in total, from 14,319 atoms $\times$ 7 $l$-components ($l_{max} = 6$)

While we use low-energy-data regimes to demonstrate improvements from Hamiltonian pretraining and investigate their origins, finetuning the pretrained HELM backbone models on a larger fraction of the energy data available in OMol25 is the next step towards achieving practically usable energy accuracies comparable with current MLIPs. Beyond that, scaling the pretraining phase to a larger number of structures may be helpful to include increased sampling of different elements. Doing so is currently constrained by memory requirements, as the number of required edge-labels and basis set size (e.g, required angular momenta) grow. Computational innovations are thus required to tackle the memory and data processing requirements of larger training labels, as well as to incorporate eigenvalue-based (Li et al., 2025) and symmetry-based (Qian et al., 2025) loss functions.

## 5 CONCLUSION

Combining the spherical harmonic representations used in equivariant GNNs with the immense volumes of information-rich training data available in the Hamiltonian matrix allows for more generalizable features to be learned from fewer atomic structure samples. Here, we introduced HELM, an efficient architecture for Hamiltonian matrix prediction, combined it with OMol_CSH_58k, a curated

Hamiltonian matrix dataset covering an unprecedented range of atomic elements and orbital interaction regimes, and demonstrated its use in learning orbital interactions and energies from a shared embedding space. HELM is state-of-the-art in Hamiltonian prediction, and using its learned features for energy prediction improves achievable test accuracy in low-data controlled experiments by the same amount as increasing training data by an order of magnitude. Overall, we establish a recipe towards integrating electronic structure data into MLIP training to learn generalizable descriptors for atomic structure-property prediction. We believe such techniques are necessary to continue to advance MLIPs.

## REPRODUCIBILITY STATEMENT

Code and the OMol_CSH_58k dataset (and corresponding test splits) will be made available after publication.

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

# APPENDIX

## A   BACKGROUND

### A.1   THE ELECTRONIC STRUCTURE PROBLEM IN DFT

In Kohn–Sham density functional theory (DFT), the electronic structure problem is recast as a set of single-particle equations for non-interacting electrons in an effective potential. When expressed in a finite atomic orbital (AO) basis $\{\chi_\mu\}$, the Kohn–Sham equations become a generalized eigenvalue problem. The molecular orbitals $\psi_i$ are expanded as

$$\psi_i(\mathbf{r}) = \sum_\mu C_{\mu i}\chi_\mu(\mathbf{r}), \tag{4}$$

where $C_{\mu i}$ are the molecular orbital coefficients. The Kohn–Sham equations in matrix form are:

$$\sum_\nu F_{\mu\nu}C_{\nu i} = \epsilon_i \sum_\nu S_{\mu\nu}C_{\nu i}, \tag{5}$$

where $F_{\mu\nu}$ is the Hamiltonian matrix, $S_{\mu\nu} = \langle \chi_\mu | \chi_\nu \rangle$ is the overlap matrix, $C_{\nu i}$ are the molecular orbital coefficients, and $\epsilon_i$ are the orbital energies. Solving **Eq. (5)** yields the molecular orbital coefficients $C$ and the orbital energies $\epsilon$, which are used to construct the electron density and compute the total energy.

### A.2   EXTRACTING ATOMIC-LEVEL OBSERVABLES FROM THE HAMILTONIAN MATRIX

Given a Hamiltonian matrix $F$ of a system of atoms, it is relatively straightforward to extract the total energy without further computation of quantum mechanical two electron (2e)-integrals or an SCF procedure. The remaining cost of evaluating the energy is thus relatively computationally inexpensive. Any number of QM packages can supply the required additional terms, such as the open-source PySCF package.

First, the density matrix is obtained from the molecular orbitals which are themselves obtained by solving the generalized eigenvalue problem $FC = SCe$ where $F$ is the Hamiltonian matrix and $S$ is the atomic orbital overlap matrix (the latter coming from PySCF) and selecting the first

$N_{\text{electron}}/2$ (assuming a restricted Kohn–Sham formalism) eigenvectors. These are the occupied molecular orbitals $C_{occ}$. Then the density matrix is:

$$P = 2C_{occ}C_{occ}^\dagger$$

The DFT energy is:

$$E = \frac{1}{2}\text{Tr}[P(H_{core} + F)] + E_{XC} - \text{Tr}[PV_{XC}] + E_{NN}$$

where $H_{core}$ is the matrix of one electron (1e) integrals, $F$ is the Hamiltonian matrix, $E_{XC}$ is the exchange-correlation energy, $V_{XC}$ is the exchange-correlation potential, and $E_{NN}$ is the nuclear-nuclear repulsion.

$$E_{NN} = \sum_{i<j}^{\text{atoms}} Z_i Z_j / r_{ij}$$

where $Z_i$ is the atomic number of atom $i$ and $r_{ij}$ is the distance between atoms $i$ and $j$. $H_{core}$, $E_{XC}$, and $V_{XC}$ can be obtained from QM codes (e.g., through PySCF). The first consists of a small number of Gaussian integrals. The latter two are obtained by evaluating the exchange-correlation functional of this density $P$ over an XC grid to obtain $E_{XC}$ and $V_{XC}$ (note that the exact exchange portion of this range-separated hybrid is already part of F and does not need to be separately computed here).

## B  HAMILTONIAN DATA PROCESSING

### B.1  PROCESSING THE HAMILTONIAN MATRIX INTO LEARNING TARGETS

Each sub-matrix block of the Hamiltonian, which represents interactions between atomic orbitals of degrees $l_1$ and $l_2$, can be decomposed and transformed into a direct sum of irreducible representations (irreps) of degree $|l_1 - l_2|$ to $l_1 + l_2$:

$$l_1 \otimes l_2 = \bigoplus_{L=|l_1-l_2|}^{l_1+l_2} L,$$

where $L$ indexes the irreps in the direct sum. This decomposition is performed using the Clebsch–Gordan coefficients, which couple the uncoupled basis of spherical harmonics $Y_{l_1 m_1}$ and $Y_{l_2 m_2}$ into the coupled basis $|LM\rangle$:

$$|LM\rangle = \sum_{m_1,m_2} C^{LM}_{l_1 m_1, l_2 m_2} |l_1 m_1\rangle |l_2 m_2\rangle,$$

In practice, the transformation from the uncoupled to the coupled basis can be expressed using Wigner 3j-symbols as:

$$C^{LM}_{l_1 m_1, l_2 m_2} = (-1)^{l_1-l_2+M}\sqrt{2L+1} \begin{pmatrix} l_1 & l_2 & L \\ m_1 & m_2 & -M \end{pmatrix}.$$

where $C^{LM}_{l_1 m_1, l_2 m_2}$ are the Clebsch–Gordan coefficients in a tensor of dimensions $(2l_1 + 1) \times (2l_2 + 1) \times (2L + 1)$, as implemented in e3nn.o3.wigner_3j.

The Hamiltonian block $H^{(l_1,l_2)}$ in the uncoupled basis, with matrix elements $H^{(l_1,l_2)}_{m_1 m_2}$, is thus transformed into the coupled basis as:

$$H^{(L)}_{MM'} = \sum_{m_1,m_2,m'_1,m'_2} C^{LM}_{l_1 m_1, l_2 m_2} H^{(l_1,l_2)}_{m_1 m_2, m'_1 m'_2} (C^{LM'}_{l_1 m'_1, l_2 m'_2})^*,$$

where $H^{(L)}$ is the block corresponding to the irrep $L$.

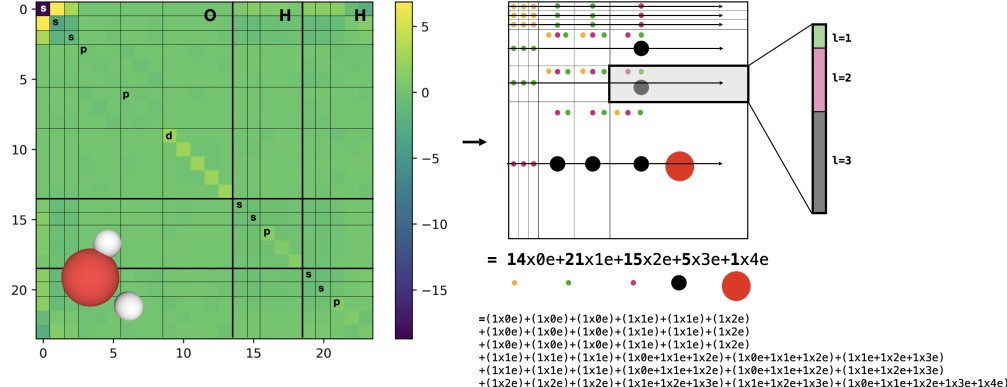

Figure 5: Dimensions of the (padded) target created for an example water molecule in a DZVP basis (taken from the MD17 dataset), and its decomposition into a direct sum of irreducible representations. The colored circles indicate the $L$s required to represent the interactions in each submatrix.

We now have irreps for every orbital interaction in $\mathbf{H}$. Next, we need to create learning targets of consistent size for every node and edge of the GNN, which involves populating the orbitally-resolved direct sum of irreps into targets of fixed size. To do this, we adopt the layout used previously by Yu et al. (2023b) and first identify a minimal 'basis' of irreps large enough to accommodate every set of orbital interactions. **Figure 5** illustrates this process for a water molecule in a DZVP basis. To account for all orbital interactions within the 9 sub-matrices of the Hamiltonian (3 node blocks, 6 edge blocks), we need a basis of $l = [0, 0, 0, 1, 1, 2]$, which corresponds to a direct sum of irreducible representations which takes the shape $[14 \times 0e + 21 \times 1e + 15 \times 2e + 5 \times 3e + 1 \times 4e]$. The orbital interaction blocks present within every node and edge can then be populated into this skeleton and processed into the coupled space with a fixed basis transformation.

Note that to accommodate all atomic interactions in the OMol_CSH_58k dataset, each set of orbital interactions (corresponding to the nodes and edges of the atomic graph) is processed into a learning target with dimension $114 \times 0e + 229 \times 1e + 239 \times 2e + 161 \times 3e + 73 \times 4e + 24 \times 5e + 4 \times 6e$. This translates to 4,096 matrix elements per node/edge.

## B.2 DECAY OF ORBITAL INTERACTIONS WITH INTER-ATOMIC DISTANCE

In **Fig. 6** we plot the elements in the Hamiltonian matrices of two representative structures from the OMol_CSH_58k dataset (one containing long-distance interactions, and another containing heavy elements), as a function of inter-atomic distance. The matrix elements are available to six decimal places, and truncation error may limit the accuracy of the last decimal place. Therefore, when processing the raw matrices to build the dataset, we trim inter-atomic interactions to a cutoff of 12Å, beyond which all $H_{ij}$ are below $1 \times 10^{-5}$. Note that this also sets a limit on the precision to which the total energy can be recomputed from this data.

## B.3 NORMALIZING SCALAR NODE BLOCK IRREPS

The Hamiltonian matrices of structures containing heavy elements often have large-magnitude trace components corresponding to the energies of the lowest-lying core electronic states. These sub-matrix traces contribute to the scalar components (irreps with angular momentum ($l = 0$)) in the targets corresponding to every node, which can be arbitrarily scaled and shifted without breaking the rotational equivariance of the data. **Figure 7** shows the set of average scalar values found in the node blocks of every atomic element for OMol_CSH_58k, averaged over the structures in the dataset. Note that the use of effective core potentials (ECPs) for the heavy elements causes a shift in the energies of the lowest $l = 0$ irreps for Rb-Xe, [Cs, Ba, La], and Hf-Rn (each of these have a different set of ECPs). For many of the heavy elements, the values range across three orders of magnitude. The corresponding scalar node values for $\nabla^2$DFT are shown on the same plot, covering a much smaller range of atomic elements and few heavy atoms (with the exception of Br).

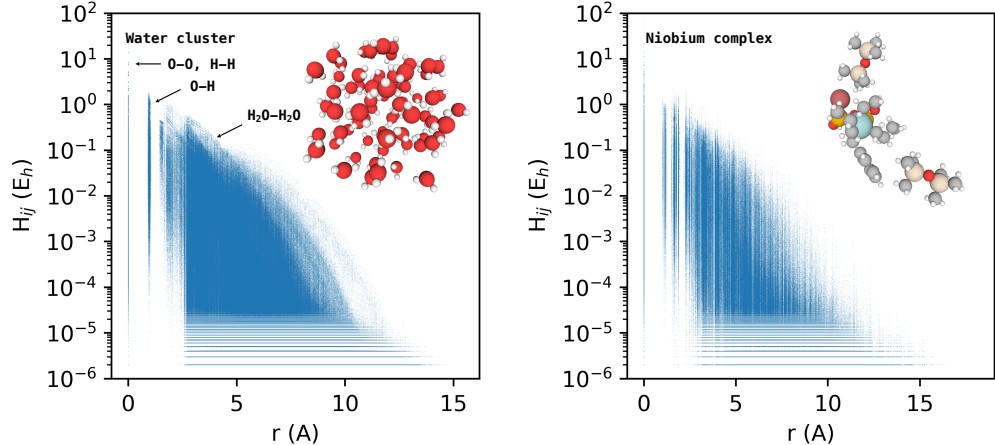

Figure 6: Magnitude of the matrix elements of $H_{ij}$ plotted as a function of interatomic distance for two representative structures from OMol_CSH_58k. The scatter points at $r = 0$ Å come from the diagonal blocks $H_{ii}$

Let $\mathbf{h}_i^{(n)} \in \mathbb{R}^d$ be the vector of scalar components (corresponding to $l = 0$ irreps) for node $n$ in structure $i$, where $d$ is the number of scalar irreps per node. For each atomic number $Z$, we collect all scalar component vectors $\mathbf{h}_i^{(n)}$ for nodes corresponding to element $Z$ across the dataset. We then compute the mean and standard deviation vectors:

$$\boldsymbol{\mu}_Z = \frac{1}{N_Z} \sum_{i,n:Z_n=Z} \mathbf{h}_i^{(n)}, \quad \boldsymbol{\sigma}_Z = \sqrt{\frac{1}{N_Z} \sum_{i,n:Z_n=Z} \left( \mathbf{h}_i^{(n)} - \boldsymbol{\mu}_Z \right)^2}$$

where $N_Z$ is the total number of scalar vectors for element $Z$ in the dataset. Any zero standard deviations, corresponding to interactions which are not sampled within a specific element's node interactions, are set to 1:

$$\sigma_{Z,j} \leftarrow \max(\sigma_{Z,j}, \epsilon), \quad \epsilon = 10^{-4}$$

Before training, the scalar components of each node $\mathbf{h}^{(n)}$ are normalized by subtracting the mean and dividing by the standard deviation corresponding to the node's element $Z$:

$$\tilde{\mathbf{h}}_j^{(n)} = \frac{h_j^{(n)} - \mu_{Z,j}}{\sigma_{Z,j}}, \quad j = 1, \ldots, d$$

where $\tilde{\mathbf{h}}^{(n)}$ is the scaled scalar vector used as the target for the equivariant graph neural network. To reconstruct the predicted Hamiltonian matrix during inference, we recover the original scale of the scalar components from the network's output $\hat{\tilde{\mathbf{h}}}^{(n)}$ with the inverse transformation:

$$\hat{h}_j^{(n)} = \hat{\tilde{h}}_j^{(n)} \cdot \sigma_{Z,j} + \mu_{Z,j}$$

In Section C.3, we include ablation studies to demonstrate the impact of this scaling procedure on the performance of HELM on the $\nabla^2$DFT Hamiltonian matrix prediction benchmark.

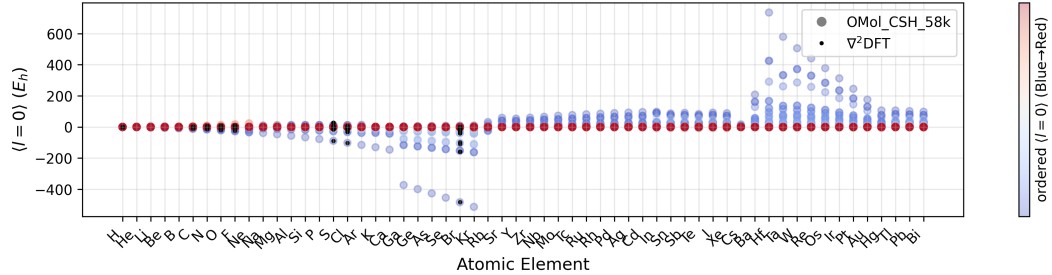

Figure 7: Scatter plot of average $l = 0$ values for every atomic element in the OMol_CSH_58k and $\nabla^2$DFT datasets, which are used in the scalar element referencing scheme. Each point corresponds to a different $l = 0$ irrep within the target for a specific atomic element, averaged over the dataset. For OMol_CSH_58k, the points are colored by their order of appearance in the target.

## C  MODEL & TRAINING

### C.1  MODEL PARAMETERS

Table 5 shows the model hyperparameters used when training each of the datasets. Note that the $l_{max}$ parameter is determined by the maximum degree of the basis required to represent all of the atomic elements in the corresponding dataset (def2-DZVP or def2-TZVPD). Scalar energy referencing is omitted for MD17 due to the lower variation in element magnitudes, and the overall simplicity of the learning task. The edge cutoff distance $r_{cut}$ is determined either by the maximum interaction distance or (in the case of OMol_CSH_58k) by the inter-atomic distance at which matrix elements are greater than $1 \times 10^{-5}$ $E_h$. Table 6 shows the number of learnable parameters available in each of HELM's components. Note that the Hamiltonian output head is designed to be relatively lightweight, in order to maintain most of the Hamiltonian-matrix-specific learnable features within the backbone embeddings.

| Data | MD17 | $\nabla^2$DFT | OMol_CSH_58k |
|---|---|---|---|
| $l_{max}$ | 4 | 4 | 6 |
| Irrep reference subtraction | None | $l = 0$ | $l = 0$ |
| Number of Message Passing layers | 3 | 3 | 3 |
| Number of embedding channels $C$ | 128 | 128 | 128 |
| Edge cutoff distance $r_{cut}$ (Å) | 16.0 | 20.0 | 12.0 |
| Gaussian cutoff (Å) | 16.0 | 20.0 | 12.0 |

Table 5: Model parameters used for MD17, $\nabla^2$DFT, and OMol_CSH_58k datasets.

| Model Component | # Parameters ($l_{max} = 4$) | # Parameters ($l_{max} = 6$) |
|---|---|---|
| Backbone - Node + Edge updates ($f^{\theta}_{[\mathbf{z}_i, \mathbf{z}_{ij}]}$) | 33,958,528 | 84,549,248 |
| Backbone - Node updates ($f^{\theta}_{[\mathbf{z}_i]}$) | 17,073,280 | 42,385,280 |
| Hamiltonian Output Head ($f^{\theta}_{\mathbf{H}}$) | 133,329 | 279,093 |
| Energy Output Head ($f^{\theta}_E$) | 5,430,017 | 13,757,185 |

Table 6: Number of learnable parameters in each model component, using the model hyperparameters specified in **Table 5** for $\nabla^2$DFT.

### C.2  TRAINING PARAMETERS

**Table 7** contains the details behind the number of train/validation/test structures used for the MD17/QM7 and $\nabla^2$DFT Hamiltonian matrix benchmarks in **Table 2**. For MD17/QM7, we follow the train/validation/test splits of QHNet Yu et al. (2023b). We limit the total training time to 24 hours on 8 GPUs (192 total node hours). The total compute cost for training thus ranges from 24 GPU hours (MD17 - Water benchmark) to 192 GPU hours (MD17 - Uracil, Ethanol, Malondialdehyde benchmarks). Loss vs. epoch plots for all four molecules are shown in **Figure 8** - note that not

all of the training converges by this point. Due to the similarity of orbital interactions between the training and test data, there is little to no overfitting observed during the training process.

For $\nabla^2$DFT, we use the train/test splits provided in Khrabrov et al. (2024), although we subtract the last 64 structures in each training split to use for validation, and use 3 message-passing layers rather than 5. Similarly to the MD17 training, the $\nabla^2$DFT benchmarks also do not converge - we stop training after 1,536 GPU hours.

To train on OMol_CSH_58k, we do not use a fixed batch size, as the large distribution of molecule sizes would introduce significant load imbalance during training (see Fig. 2). Instead, we batch the data by 'binning' molecules to roughly 22k graph edges per bin, which typically translates to between 1 and ∼83 molecules. Doing this allows for better load balancing during distributed training, since the time to process a molecule is determined almost entirely by the number of edges to which its graph representation is connected (which determines the number of messages processed within each layer of the model). When training is distributed on 64 processes, we then have roughly 1,408k graph edges per batch.

| | **MD17** | | | | **$\nabla^2$DFT** | | | | | | **OMol_CSH_58k** |
| | | | | | **Train** | | | **Test** | | | |
| | Water | Uracil | Ethanol | Malondialdehyde | 2k | 5k | 10k | 2k | 5k | 10k | |
|---|---|---|---|---|---|---|---|---|---|---|---|
| # Train | 500 | 25k | 25k | 25k | 12,145 | 28,362 | 49,725 | – | – | – | 56k |
| # Val. | 500 | 500 | 500 | 500 | 64 | 64 | 64 | – | – | – | 64 |
| # Test | 3900 | 1478 | 4.5k | 4.5k | – | – | – | 2,774 | 5,634 | 9,532 | 5k/1k |
| Batch size | 5 | 8 | 8 | 8 | 64 | 64 | 64 | 64 | 64 | 64 | ∼1,408k edges |
| Scheduler | | ReduceLRonPlateau | | | | ReduceLRonPlateau | | | – | | CosineAnnealingLR |
| Initial LR | | $1 \times 10^{-4}$ | | | | $1 \times 10^{-4}$ | | | – | | $1 \times 10^{-3}$ |
| Optimizer | | Adam | | | | Adam | | | – | | AdamW |
| Precision | | double | | | | | double | | – | | single |

Table 7: Hamiltonian matrix training parameters for MD17, $\nabla^2$DFT, and OMol_CSH_58k datasets. MD17 is shown per molecule, and $\nabla^2$DFT shows parameters for multiple splits. MD17 splits are taken from Yu et al. (2023b).

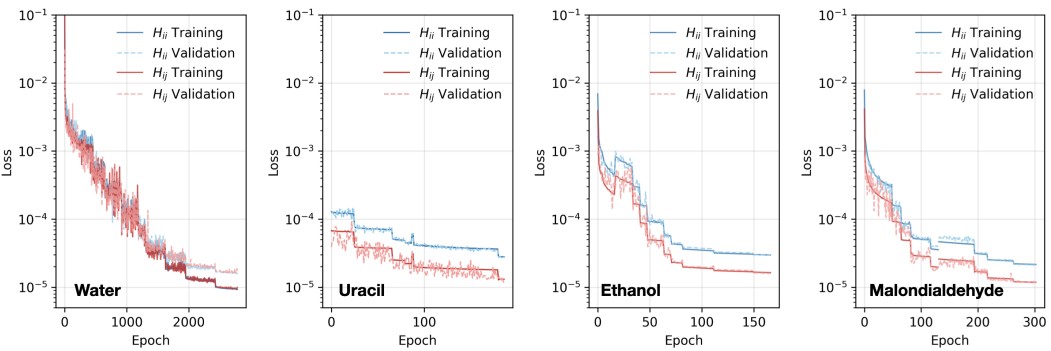

Figure 8: Loss vs. Epochs for the four molecules in the MD17 database.

| **Data** | | **$\nabla^2$DFT** | | | | | **OMol_CSH_58k** |
| | | **Train** | | | **Test** | | |
| | 2k | 5k | 10k | 2k | 5k | 10k | |
|---|---|---|---|---|---|---|---|
| # Train | 8,097 | 18,908 | 33,150 | – | – | – | 56k |
| # Validation | 640 | 640 | 640 | – | – | – | 64 |
| # Test | – | – | – | 2,774 | 5,634 | 9,532 | – |
| Batch size | 640 | 640 | 640 | – | – | – | ∼1,408k edges |
| Number of Epochs | 10,000 | 5,000 | 3,000 | – | – | – | 700 |
| Scheduler | | CosineAnnealingLR | | | | | CosineAnnealingLR |
| Initial learning rate | | $1 \times 10^{-6}$ | | | | | $1 \times 10^{-4}$ |
| Minimum learning rate | | $1 \times 10^{-8}$ | | | | | $1 \times 10^{-8}$ |
| Optimizer | | Adam | | | | | AdamW |
| Precision | | double | | | | | single |

Table 8: Energy training parameters for $\nabla^2$DFT and OMol_CSH_58k datasets.

**Table 8** contains the training parameters used for the energy training results presented in **Table 3** and **Table 4**. For $\nabla^2$DFT we train on 2/3 of the molecules in the dataset, and use the remaining 1/3 for validation. We use the same root-MSE+MSE loss as for Hamiltonian prediction for consistency.

### C.3 Ablations on node-parity expansion and $l = 0$ scaling

Here we preform ablation studies to understand the impact of two specific features of HELM: (1) the node parity expansion blocks described in **Section 3** ('Hamiltonian output head'), and (2) the $l = 0$ component scaling described in **Section B.3**. In **Table 9** we report the results of these ablations using the $\nabla^2$DFT 2k train/test splits. All of the training here was done with a fixed number of epochs (500) and a cosine scheduler. We use a fixed number of epochs specifically to compare between these different settings - this is generally not long enough to converge the training (which was done using a ReduceLRonPlateau scheduler for the results in **Table 2**), so we also report the error on the train split.

| $l = 0$ **Scaling** | **Node-Parity Expansion** | $H^{err}$ - **train** | $E^{err}$ - **train** | $H^{err}$ - **test** | $E^{err}$ - **test** |
|---|---|---|---|---|---|
| ✗ | ✗ | 201.21 | 169,026.08 | 324.35 | 329,923.53 |
| ✗ | ✓ | 95.73 | 1,163.54 | 101.73 | 1,530.83 |
| ✓ | ✗ | 115.35 | 837.07 | 120.18 | 973.47 |
| ✓ | ✓ | **92.00** | **540.19** | **96.27** | **706.39** |

Table 9: Effect of node-parity reduction and $l = 0$ referencing scheme on matrix element error, evaluated using the $\nabla^2$DFT 2k train/test splits. $H^{err}$ is in units of $\times 10^{-6} E_h$, while $E^{err}$ is in units of meV. The best case performance in each column is in bold.

The node parity expansion block only impacts the error within diagonal blocks of the Hamiltonian. These diagonal blocks contain the largest-magnitude elements (resulting from nearsightedness of orbital interactions), and contribute disproportionately towards the eigenvalue error and total energy prediction. Therefore, improving the loss over diagonal orbital blocks has a significant impact on the total energy computed from the matrices, bringing the error down from 329,923 meV to 1,530 meV. The $l = 0$ component scaling procedure serves to reduce the numerical range of the data from the order of $\sim 100$ to the order of $\sim 1$ (see **Fig. 7** in **Section B.3**), thus making it easier to learn with a properly-normalized model. This brings the energy error down further to 706 meV. Together, these two architectural choices contribute significantly towards the improved performance of HELM over other models in **Table 2**.

## D Additional results

### D.1 Hamiltonian matrix error analysis - $\nabla^2$DFT

In **Fig. 9** we show the correlation between matrix element errors and eigenvalue errors, collected over all three of the considered test splits in $\nabla^2$DFT. In general, a weak correlation exists between matrix element errors and resulting eigenvalue errors. We also note that larger molecules tend to have lower matrix element and total energy errors, likely stemming from increased sparsity in $\mathbf{H}$.

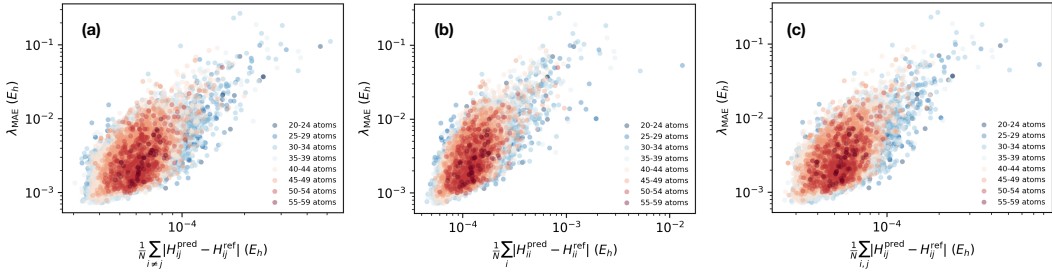

Figure 9: Correlation between matrix element errors (separated into (a) edge block errors, (b) node block errors and (c) total matrix elements) and resulting MAE errors over the occupied energy eigenvalues ($\lambda_{MAE}$) of the predicted and reference matrices, shown for all the data over the 2k, 5k, and 10k test splits of $\nabla^2$DFT.

In Fig. 10 we additionally examine the relationship between matrix element errors and the error in the total energy computed from the predicted $\mathbf{H}$. Most errors are on the order of $1 \times 10^{-2} E_h$ (272.114 meV), however several outliers can be seen with total energy errors of $> 1$. When inspecting these cases manually, we found that they occur from cases where nearly-degenerate eigenvalues (to within the prediction error) cause inconsistent electron occupation patterns between the reference and predicted matrices, leading to different density matrices computed from them, and thus to different total energies. To address this, we filtered out linearly dependent atomic orbitals by first removing eigenvalues of the computed overlap ($\mathbf{S}$) matrix to within $1 \times 10^{-3}$, followed by manually filtering out inconsistent occupations. We note that the removal of linear dependencies is performed internally within many DFT codes, and typically makes a negligible difference on the value of the computed total energy.

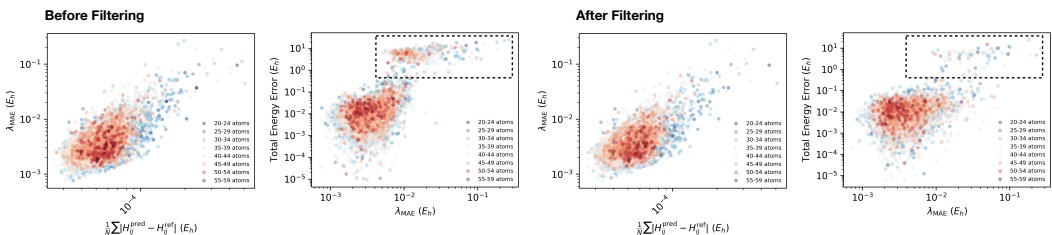

Figure 10: Correlation between matrix element errors, occupied energy eigenvalue errors ($\lambda_{MAE}$) and total energy computed from the predicted and reference matrices, shown for all the data over the 2k, 5k, and 10k test splits of $\nabla^2$DFT. The plots are shown before/after filtering the eigenvalues of $\mathbf{S}$ to under $1 \times 10^{-3}$. The black dashed boxes indicate outliers stemming from different occupation, which are reduced after the filtering process.

To understand how these matrix errors are distributed as a function of inter-atomic distance, in **Fig. 11** we plot the predicted matrix elements and labels for three representative test conformers in the 10k set. The data is plotted in the coupled basis - before the reverse-transformation required to reconstruct the Hamiltonian - in order to additionally differentiate between irreps of different degree $l$. An overall matrix element error of $\sim 60 \mu E_h$ (**Table 2**) results in the decay in magnitude of orbital interactions with inter-atomic distance being captured up to 15 Å. The highest remaining relative errors are found in the longer-range orbital interactions, where the magnitude of the matrix elements drops below $100 \mu E_h$. These long-range interactions are not found in the small molecules of MD17, where the largest interatomic distances are typically less than 5 Å, which partially contributes to the large difference in matrix element errors between datasets. We also note that the contributions of different degree $l$ to the matrix elements are learned simultaneously, so the model does not prioritize lower or higher angular degrees.

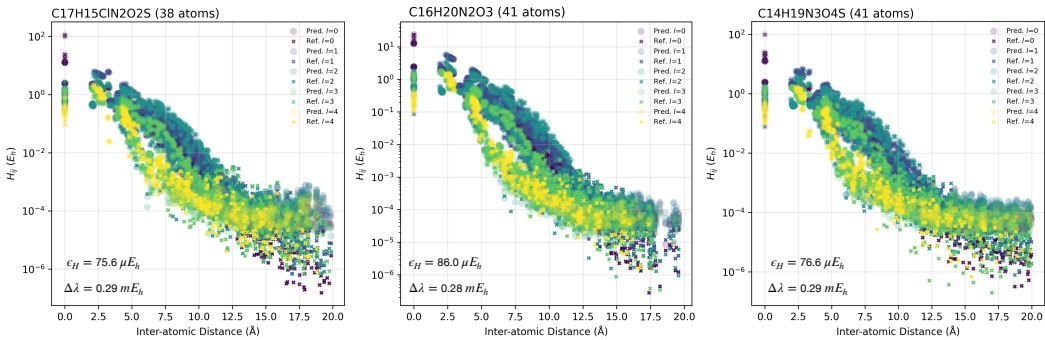

Figure 11: Decomposition of the matrix elements of $\mathbf{H^{ref}}$ and $\mathbf{H^{pred}}$ by interatomic distance and angular degree $l$, shown for three molecules in $\nabla^2$DFT. Points at an inter-atomic distance of 0.0 Å correspond to elements within the diagonal Hamiltonian submatrices/node outputs. The inset shows the connectivity of graph representation of each molecule with an $r_{cut}$ of 20 Å.

## D.2 OMOL_CSH_58K HAMILTONIAN PREDICTIONS

In Fig. 12, we show computed Hamiltonian matrices with HELM for the first few structures in the OMol_all_5k test split. We also show heatmaps of the reference Hamiltonian matrices, as well as the energy eigenvalues computed between them. For numerical stability, the eigenvalues of the overlap matrix used to solve the generalized eigenvalue problem with both reference and predicted Hamiltonian matrices were filtered to $1 \times 10^{-1} E_h$. In general, we reproduce the matrix elements to $\sim 2 \times 10^{-3} E_h$ (see Table 4), which in practice yields visually indistinguishable matrices and eigenvalues for the $\mathbf{H}$ in this test split. Deviations from the reference energy eigenvalues are typically concentrated around band edges.

To further analyze the utility of the predicted matrices, we use them to re-compute the electron density with PySCF (see Fig. 13). Visually, the electron density isosurfaces computed for all three datasets are nearly indistinguishable from those computed directly from the reference Hamiltonian matrices.

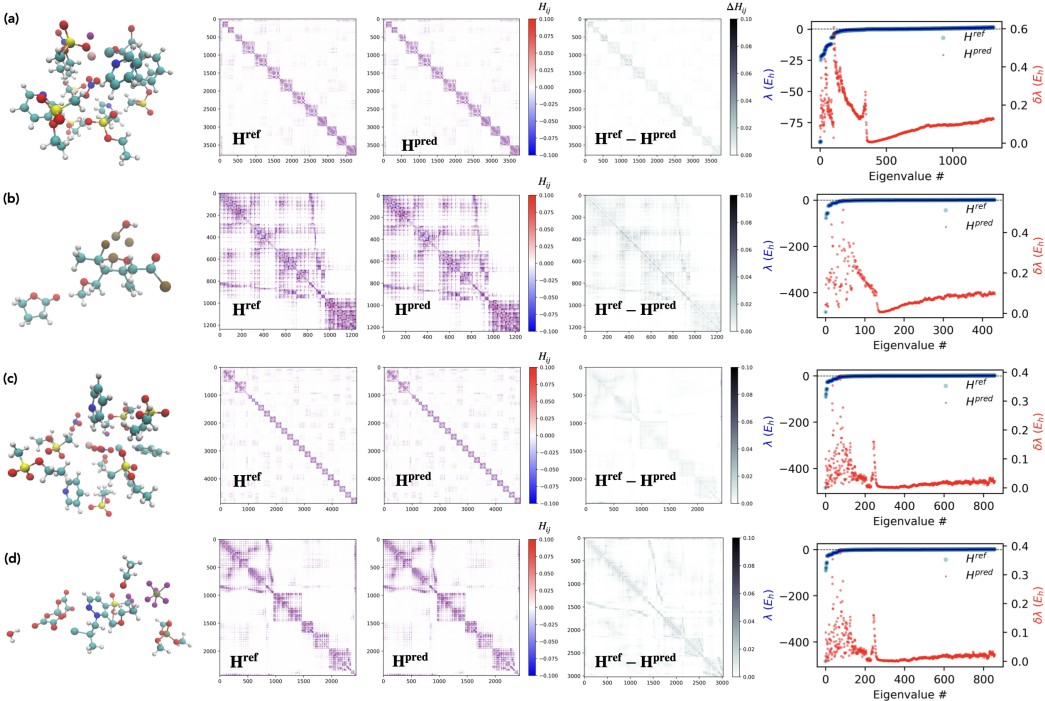

Figure 12: Predicted Hamiltonian matrices from the first four atomic structures in the OMol_all_5k test split, showing heatmaps of the label and predicted $\mathbf{H}$ (bounded to a color range of [-0.1, +0.1] to better distinguish patterns in inter-orbital interactions), the element-wise error between them, as well as a comparison of the energy eigenvalues computed from them.

## D.3 ENERGY PREDICTION ON DIFFERENT CONFORMERS SPLITS - $\nabla^2$DFT

In Table 3, we evaluated energy errors on matching 2k, 5k, and 10k conformer train/test splits in $\nabla^2$DFT, assessing generalization to different conformers of molecules present in the training set. Hamiltonian pretraining significantly reduced energy prediction errors. In Table 10, we extend this analysis to test generalization to unseen molecules by evaluating each model, trained on 2k, 5k, or 10k splits, across all test splits. Hamiltonian pretraining consistently outperforms direct training on unseen molecules, even when a finetuned model trained on the 2k split is tested on the additional 8k molecules in the 10k split (8.62 meV/atom) compared to an energy-direct model trained on the 10k split and tested on 2k (11.0).

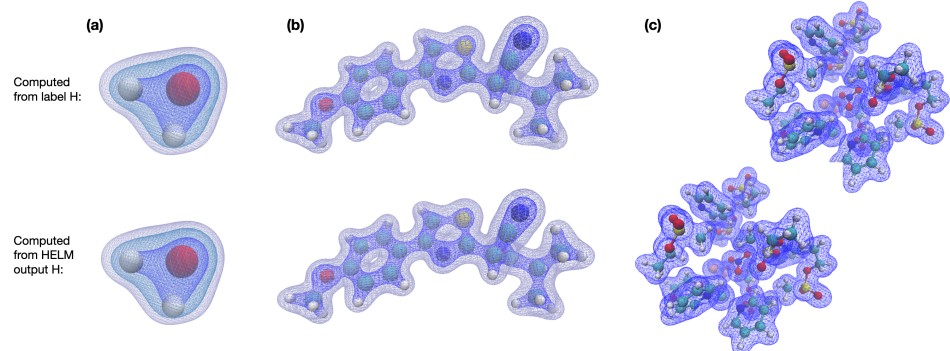

Figure 13: Isosurfaces of electron density visualized for a random molecule taken from the (a) QM7 (isovalues: 0.1, 0.2, 0.5 e$^-$/$a_0^3$), (b) $\nabla^2$DFT Test-2k (isovalues: 0.05, 0.2 e$^-$/$a_0^3$) and (c) OMol_CSH_58k (isovalues: 0.03 e$^-$/$a_0^3$) test splits. In each case, the upper row shows the electron density plotted with the label Hamiltonian as input, and the bottom row shows the one predicted from HELM's output.

| | **Train** | Test-2k (2,774) | Test-5k (5,634) | Test-10k (9,532) |
|---|---|---|---|---|
| **Direct** | | | | |
| Train-2k (8,097) | 2.68 | 21.7 | 20.3 | 23.4 |
| Train-5k (18,908) | 6.08 | 11.9 | 15.6 | 14.2 |
| Train-10k (33,150) | 7.00 | 11.0 | 10.2 | 13.0 |
| **Pretrained-frozen** | | | | |
| Train-2k (8,097) | 5.66 | 9.00 | 10.1 | 11.0 |
| Train-5k (18,908) | 6.56 | 7.61 | 8.02 | 8.18 |
| Train-10k (33,150) | 5.75 | 6.84 | 6.42 | 6.93 |
| **Finetuned** | | | | |
| Train-2k (8,097) | 3.74 | 7.32 | 7.62 | 8.62 |
| Train-5k (18,908) | 3.71 | 4.29 | 5.22 | 4.87 |
| Train-10k (33,150) | 4.20 | 4.80 | 4.49 | 5.05 |

Table 10: MAE for the direct, pretrained-frozen, and finetuned HELM models trained/evaluated on the different conformers splits of $\nabla^2$DFT. In brackets are the number of molecules used for training/testing. All units are in **meV/atom**.

## D.4 EXTENDED UMAP EMBEDDING ANALYSIS

In Fig. 14 and Fig. 15, we show an extended version of Fig. 4 for both the $\nabla^2$DFT and OMol_CSH_58k datasets. We extract embeddings from 100 molecules of the train-2k split for the former, and 250 molecules for the latter. Clusters from the "direct" model indicate that the ability to distinguish hydrogen and carbons atoms has been learned, consistent with the ubiquity of these elements in the training data. After Hamiltonian pretraining, a greater variety of features for different atomic elements can be distinguished. The remaining similarities occur between pairs such as C-H and O-F, which may have more structurally similar atomic environments within the training data.

In the second row of each figure, we show the UMAP visualization, now colored by angular degree $l$. The embeddings after Hamiltonian pretraining on OMol_CSH_58k exhibit greater separation in the space of irreps of different degree $l$. Our results are thus consistent with the observations of Lee et al. (2024) and Miedaner et al. (2025), where training on rank-2 targets (molecular hyperpolarizabilities) indicated better utilization of higher-degree irreps than training on scalar (rank-0) targets (energies). Extending this idea, the Hamiltonian matrix can be thought of as containing the highest-rank tensors required to encode the properties of a system of atoms in a given atomic orbital basis. Training to learn these matrix elements results in the model maximally leveraging different components of the spherical harmonic embeddings to extract structural descriptors from the training data.

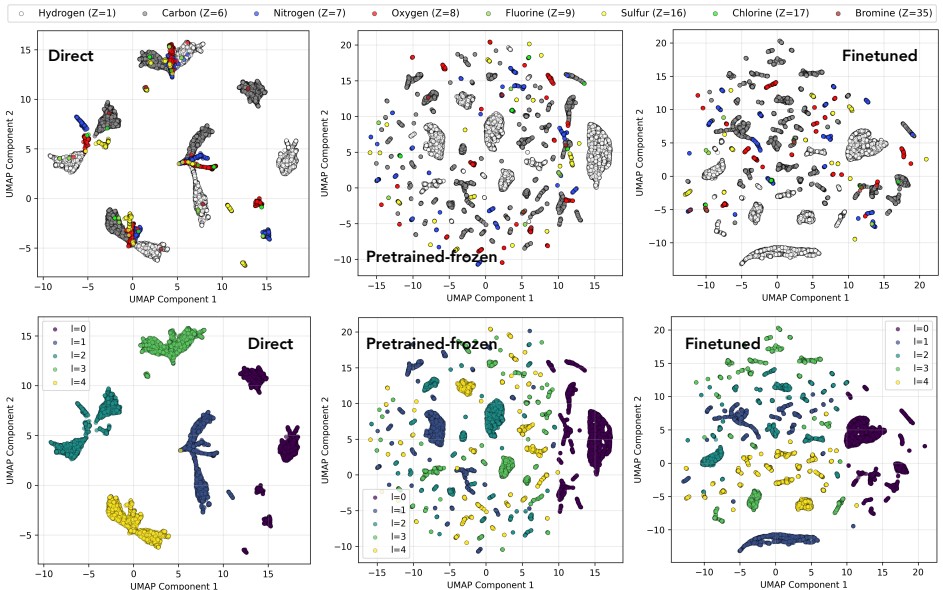

Figure 14: UMAP node embeddings for the first 100 molecules of the $\nabla^2$**DFT** train-2k split, colored by (top) atomic element and (bottom) angular degree $l$. There are 19,040 data points extracted in total, from 3,808 atoms $\times$ 5 l-components ($l_{max} = 4$).

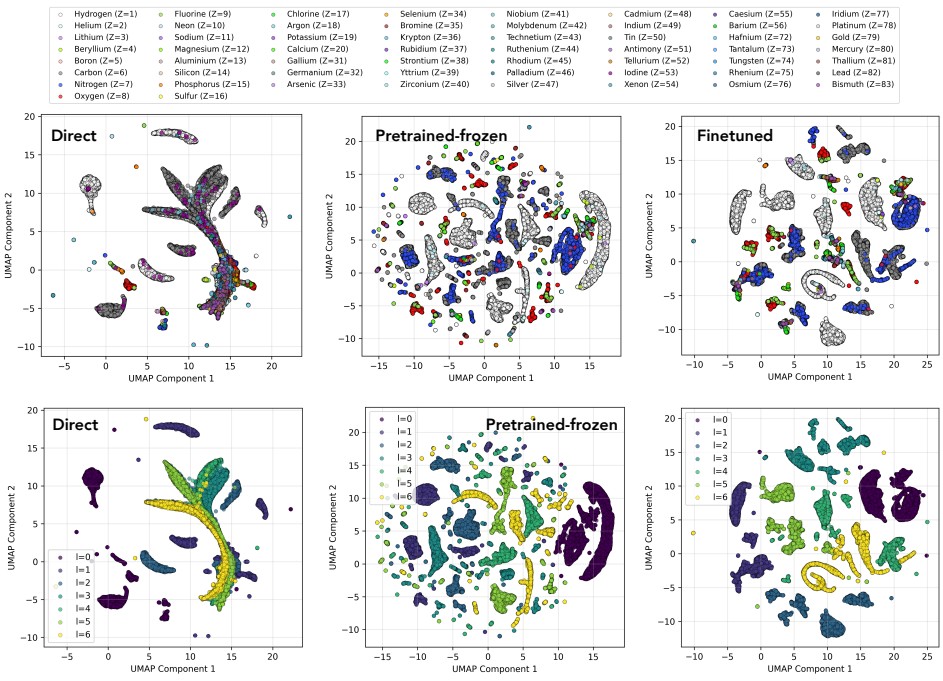

Figure 15: UMAP node embeddings for **OMol_CSH_58k**. The lower row of visualizations show the same data as in **Fig. 4**, but now colored by angular degree $l$.

# E    USE OF LLMS

We made use of LLMs to refine the LaTeX and math formatting within this document. No new ideas or text were generated by LLMs.

