# OpenReview forum: "Learning from the Electronic Structure of Molecules across the Periodic Table"
_ICLR.cc/2026/Conference — ICLR 2026 Poster_

### Official Review · Reviewer_QD9Z · 2025-10-23

**Soundness:** 3
**Presentation:** 3
**Contribution:** 2
**Rating:** 4
**Confidence:** 5

**Summary:**

This paper introduces HELM, a model that predicts Hamiltonian matrices for molecules, enabling the integration of orbital interaction into interatomic potential training. Accompanying HELM, the authors release OMolCS-H58k—a large-scale Hamiltonian dataset featuring 58 elements and molecules up to 150 atoms. They also propose “Hamiltonian pretraining,” a method that leverages Hamiltonian-derived embeddings to enhance energy prediction accuracy.

**Strengths:**

* The paper is well-structured and clearly written, providing a concise and accessible overview of the problem and the proposed solution.
* The authors conduct a thorough analysis, enhancing understanding through techniques such as UMAP visualization.
* Moreover, they introduce a new dataset of Hamiltonians, which is a valuable contribution to the open-source community in this field.

**Weaknesses:**

I do not find any obvious weaknesses in this paper. I believe that many researchers working in the areas of Hamiltonian prediction and machine learning force fields may have had similar ideas, so the concept itself is reasonable, and the experimental results appear solid. However, I do have some questions about the paper—please see Questions. If the authors address my concerns, I will consider increasing my score.

**Questions:**

* In the Introduction, the paper states: “For every system of N atoms, there is 1 energy label, O($N$) force labels, and O($N^2$) labels in the Hamiltonian matrix.” However, the number of labels in the Hamiltonian matrix should be the total number of orbitals, not the number of atoms.
* In "Learning orbital interactions" of Background and Previous Work, both the initial models and following work are cited as [1]. Is this expected?
* In Fig. 2(d), the “Element–element interaction” is shown. How are these numerical values computed?
* In Fig. 3(c), are the results shown the same as those reported in Table 4? In Fig. 3(c), the pretrained-frozen model appears to achieve better validation results, whereas Table 4 shows that the finetuned model performs better. These seem inconsistent. I guess that this be due to the loss functions being different (MSE in the figure vs. MAE in the table)? In the section Loss and scalar referencing, the paper describes the loss function used for Hamiltonian prediction, but since the model is also trained to predict energy, I could not find a clear explanation of the loss used for energy prediction. Could the authors clarify this?
* The work applies a cutoff to the interactions. Appendix B.2 mentions: “Note that this also sets a limit on the precision to which the total energy can be recomputed from this data.” Could a quantitative error analysis be provided here? How much error does this cutoff introduce in downstream tasks?
* In Appendix D.1, Fig. 9(c) shows a trend that appears inconsistent with Fig. 6 in [2] and Fig. 1 in [3], both of which display a divergent trend. Could the authors explain the source of this discrepancy?

[1] Gong X, Li H, Zou N, et al. General framework for E (3)-equivariant neural network representation of density functional theory Hamiltonian[J]. Nature Communications, 2023, 14(1): 2848.

[2] Wang Z, Liu C, Zou N, et al. Infusing self-consistency into density functional theory hamiltonian prediction via deep equilibrium models[J]. Advances in Neural Information Processing Systems, 2024, 37: 89652-89681.

[3] Li Y, Xia Z, Huang L, et al. Enhancing the scalability and applicability of kohn-sham hamiltonians for molecular systems[J]. arXiv preprint arXiv:2502.19227, 2025.

---

> ### Author Response · Authors · 2025-11-25
>
> Thank you for reviewing our work in depth. Pretraining on electronic structure data indeed seemed like a logical step to unlock the next leap in the capabilities of ML molecular property models, and we are glad to hear that other researchers were thinking in a similar direction. We hope that the release of HELM and the curated OMol\_CSH\_58k dataset will simplify the pretraining process for others.
>
> Questions:
>
> 1. For a system of $N$ atoms in a specified orbital basis, the number of orbitals in the system is approximately $cN$, where $c$ is the number of orbitals per atom (roughly, changes per atom type). $c$ does not change when the size of the system is increased, so the scaling behavior is $\mathcal{O}(N^2)$. Another way to interpret this comparison is by the number of GNN components for which there is unique data to map to. Then, for a graph of $N$ nodes, there is one energy label, $\mathcal{O}(N)$ force labels, and $\mathcal{O}(N^2)$ Hamiltonian matrix labels.
>
> 2. We intended to only cite it the first time (corrected) - thank you!
>
> 3. The values in this heatmap correspond to the log of the number of times atoms of element X and Y appear within 12.0 A (cutoff) of each other, across all molecules in the dataset. Previously we only mentioned the log in the subfigure, but have now also added it in the caption: ``...log of..."
>
> 4. Yes, the results shown are the same. We may have created some confusion as Table 4 is the `transpose' of Table 3 (for space reasons). In Table 4, the pretrained case achieves lower losses than the finetuned one, consistent with Fig. 3(c):
>
> OMol\_common\_1k: Pretrained (2119.76) $<$  Finetuned (2533.22)
>
> OMol\_all\_5k: Pretrained (3255.69) $<$ Finetuned (3926.12)
>
> For energy training, we use the same loss function as for Hamiltonian prediction for consistency (MSE+RMSE), and apply element-wise references to normalize the data [1]. We have now added a sentence stating this to Appendix C.2.
> [1] ACS Catal. 2023, 13, 5, 3066–3084, preprint: https://arxiv.org/abs/2206.08917 (page 7 of the supplementary or page 34 of the preprint)
>
> 5. A small clarification: in Appendix B.2 we mention truncation error in the raw data (from the ORCA calculation output files). The precision at which the truncation error becomes evident coincides with the maximum magnitude of orbital interactions at roughly 12 A (see Fig. 6), so this became a natural point at which to set the interaction cutoff.
>
> The question becomes: how does this truncation error in the printed ORCA Hamiltonian matrix elements limit the accuracy to which the energy can be computed directly from them? We can approximate this by comparing (1) energy computed from the OMol\_CSH\_58k label Hamiltonian (using the same functional and grid) with (2) the energy label for the same molecule from ORCA. For a sample water molecule:
>
> (1):  -76.4242869055408 $E_h$
>
> (2):  -76.4242861700105 $E_h$
>
> The difference is 0.02 meV.
>
> 6. The trend in Fig. 6 in Ref [2] is not clearly divergent within the limited range shown, so we focus on Fig. 1 in Ref [3].
>
> Our model outputs are not directly comparable to that of Ref [3], Fig. 1, as (1) we scale out element-wise reference values from the l=0 components of the nodes (section B.3). This decreases the numerical range of diagonal elements from the order of $\sim$100 to the order of $\sim$1 (see Fig. 7 in Section B.3), bringing them to a similar magnitude as many of the off-diagonal elements. We also (2) incorporate parity reduction blocks for the nodes, which results in the predicted diagonal blocks being symmetric.
>
> We did some additional ablations to illustrate the impact of (1)-(2). The results are summarized below, along with data from Ref [3] on their PubChemQH dataset. This is not a direct comparison, it only serves to show that, when omitting the $l=0$ scaling and node parity reduction from HELM, we see a large deviation in the difference between matrix element error and total energy error, consistent with what is observed in Ref [3] without the WALoss. Our response to Reviewer atd4 includes a further breakdown.
>
> | Model               | Dataset    | Basis      | Node-Parity | $l=0$ scaling | $H^{err}$ ($\mu E_h$) | Energy (meV)  |
> |---------------------|------------|------------|-------------|---------------|-----------|----------|
> | WANet w/o WALoss    | PubChemQH  | def2-TZVP  | -           | -             | 999.8     | 2,757,060|
> | WANet w/ WALoss     | PubChemQH  | def2-TZVP  | -           | -             | 756.0     | 2,046    |
> | HELM                | $\nabla^2$DFT | def2-SVP  | ×           | ×             | 324.4     | 329,923  |
> | HELM                | $\nabla^2$DFT | def2-SVP  | ✓           | ✓             | 96.3      | 706      |
>
> The data from WANet is converted from values in Table 1 of Ref [3]. Ablations on $\nabla^2$DFT were trained on the 2k train split with a fixed number of 500 epochs with a cosine scheduler (thus differing from Table 2 of the paper), and evaluated on the 2k test split.

---

> > ### Comment · Reviewer_QD9Z · 2025-11-25
> >
> > Thanks for the authors' response. Most of my concerns are addressed, so I will increase my score.

---

### Official Review · Reviewer_RqiD · 2025-10-30

**Soundness:** 2
**Presentation:** 2
**Contribution:** 2
**Rating:** 6
**Confidence:** 5

**Summary:**

This paper introduces HELM (Hamiltonian Electronic Learning Model), an SO(3)-equivariant graph neural network that jointly learns to predict Kohn-Sham Hamiltonians and molecular energies. The key innovation is leveraging the rich electronic structure information encoded in Hamiltonian matrices—routinely computed but typically discarded during DFT calculations—as a pretraining task to improve machine learning interatomic potentials (MLIPs). The authors construct OMol-CSH-58k, a new dataset of ~56.7k closed-shell molecules covering 58 elements with up to 150 atoms, and demonstrate that Hamiltonian pretraining yields substantial improvements in energy prediction accuracy, particularly in low-data regimes.

**Strengths:**

*  The paper makes a good case for exploiting Hamiltonian matrices as supervisory signals. The connection between electronic structure and atomistic properties is conceptually novel, and the two-head design bridges DFT and MLIP domains.
*  OMol-CSH-58k addresses a  gap in existing Hamiltonian datasets by covering 58 elements (vs. 8-10 in prior work), molecules up to 150 atoms, and using the larger def2-TZVPD basis. The dataset composition analysis (Fig. 2, Table 1) is thorough.
*  HELM achieves ~60 µEh error on ∇²DFT compared to 520-20700 µEh for prior methods (Table 2). The margin is meaningful  The analysis connecting matrix errors to eigenvalue and energy errors (App. D.1-D.2) strengthens the claims.
*  The pretraining protocol shows energy MAE improvements, especially on ∇²DFT with limited training data (up to ~2× reduction, Table 3). The frozen vs. finetuned head comparisons are informative, and UMAP visualizations (Fig. 4) support the claim of learning better representations.
*  Distance-decay analysis justifying the 12Å cutoff (App. B.2, Fig. 6), element-referenced scalar normalization (App. B.3), and batching strategies for handling size heterogeneity (Table 7) demonstrate careful implementation.

**Weaknesses:**

* There is a concerning mismatch between the exchange–correlation (XC) functionals used for generating the training data and those used for reconstructing energies from predicted Hamiltonians. Appendix A.2 reports ωB97M-V for energy reconstruction, whereas the ∇²DFT data rely on ωB97X-D/def2-SVP (Table 2). Such inconsistency may induce systematic biases and plausibly account for outliers observed in Fig. 10. The paper should clearly specify: (i) the XC functional used for energy reconstruction in each dataset, (ii) whether this matches the functional that produced the reference data, and (iii) the quantitative impact of any mismatch.

* The model integrates several design elements—SO(2) convolutions, gated nonlinearities per irrep, parity-aware constraints, and element-referenced normalization—but their individual contributions are not analyzed. It remains unclear whether the observed gains stem primarily from architectural innovations or auxiliary components such as normalization or loss design. A systematic ablation table is necessary to disentangle these effects and to provide guidance for future model development.
* Table 4 and Fig. 3c show cases where fine-tuning on OMol leads to degraded performance relative to the frozen pretrained head, suggesting overfitting. While the authors attribute this to high elemental diversity, the explanation remains speculative. A more rigorous analysis should explore regularization strategies (e.g., layer freezing, dropout, weight-decay sweeps, early stopping) and report per-element error breakdowns to identify whether rare elements disproportionately drive the issue.
* Energy errors are inconsistently reported—sometimes per-molecule (Table 3) and sometimes per-atom (Table 9)—without clear indication. The notation alternates between H and F (Fock) across sections. Units and symbols should be standardized throughout (e.g., meV / atom) and summarized in a concise notation table for clarity.

**Questions:**

1. Can a model pretrained on def2-SVP (∇²DFT) transfer to def2-TZVPD (OMol) with light head retraining, or vice versa?
2. Can you also add a force head to your model to see pretraining on H helps with force prediction?
3. What modifications would HELM require for open-shell/charged systems?
4. Please report detailed computational performance metrics, including:  (i) Wall-clock time per epoch for ∇²DFT-2k on V100 and A100 GPUs  (ii) Peak GPU memory usage as a function of $l_{\\text{max}}$ (iii) Inference time per molecule across different molecular size ranges

---

> ### Author Response · Authors · 2025-11-25
> **Discussion on mentioned weaknesses**
>
> Thank you for reviewing our paper! We have (1) clarified the ambiguity about the functional used to compute the energy from the Hamiltonian matrices of MD17 and $\nabla^2$DFT, (2) added ablation studies on model components to explain the origin of Hamiltonian prediction improvements, and (3) added the requested data on computational cost of training and inference in the computationally-heaviest Hamiltonian training/inference mode of HELM. We look forward to further discussion and are happy to provide additional data or clarifications.
>
> [Weakness-1]: The functional used to compute the energy is consistent with the one which was used to generate the data, in all three cases (MD17, $\nabla^2$DFT, and OMol\_CSH\_58k). The functional mentioned in the text of Appendix A.2, however, was specific to OMol\_CSH\_58k. We apologize for this confusion (now clarified in the paper). In summary:
>
> | Dataset | Basis | Functional | Grid |
> | :--- | :--- | :--- | :--- |
> | OMol\_CSH\_58k | def2-tzvpd | wb97m-v | atom\_grid=(99,590), nlc\_grid=(50,194) |
> | MD17 | def2-svp | pbe | atom\_grid=(99,590) |
> | $\nabla^2\text{DFT}$ | def2-svp | wb97x-d | atom\_grid=(75,302) |
>
>
> [Weakness-2]: The primary architectural innovations over other models in Table 2 are:
>
> (1) eSCN convolutions (first introduced in Ref [1]) to treat the $l_{max}$ of 4 and 6 required by the Hamiltonian datasets
>
> (2) Coupled node/edge updates (introduced for the purpose of downstream property finetuning)
>
> (3) Symmetry-preserving data scaling
>
> (4) Node-parity expansion blocks
>
> We acknowledge that we omitted thorough ablations of (3) and (4), which were included specifically to improve training on Hamiltonian data. We have now included ablations on these two additions, which are summarized below. All of these were done on the $\nabla^2$DFT 2k train/test splits, with a fixed number of epochs (500) and a cosine scheduler for fair comparison between different settings. Note that this is not long enough to fully converge training (which was done using a ReduceLRonPlateau scheduler for the results in Table 2 of the paper, and converged at 1200 epochs for the 2k split).
>
> | $l=0$ Scaling | Node-Parity Expansion | $H^{err}$ - train ($\times 10^{-6} E_h$) | $E^{err}$ - train (meV) | $H^{err}$ - test ($\times 10^{-6} E_h$) | $E^{err}$ - test (meV) |
> |---------------|-----------------------|-------------------|-------------------|------------------|------------------|
> | ×             | ×                     | 201.21            | 169,026.08        | 324.35           | 329,923.53       |
> | ×             | ✓                     | 95.73             | 1,163.54          | 101.73           | 1,530.83         |
> | ✓             | ×                     | 115.35            | 837.07            | 120.18           | 973.47           |
> | ✓             | ✓                     | 92.00             | 540.19            | 96.27            | 706.39           |
>
> Overall, we found that these innovations which take advantage of parity-symmetry in the data and preservation of rotational properties of the $l=0$ components under scaling, combined with careful model implementation and training pipelines, allowed us to significantly push the state of the art in Hamiltonian prediction, and led to the results of Table 2. In our response to Reviewer atd4, we have added further explanations behind these ablations and why they have such a high impact on the error. Our response to Reviewer QD9Z's last question also includes a comparison of the before/after with Ref [2], who observed similarly high total energy values in their unoptimized case, consistent with our observations.
>
> [1] Saro Passaro and C. Lawrence Zitnick. Reducing SO(3) convolutions to SO(2) for efficient equivariant GNNs, 2023. URL https://arxiv.org/abs/2302.03655.
>
> [Weakness 3]: Due to the complexity of re-training these models under the given scenarios, we are currently unable to thoroughly explore the suggested regularization strategies. But we find these interesting propositions for future work.
>
> [Weakness 4]: The caption of Table 9 in the Appendix indicates, ``All units are in meV/atom" in bold font, because it is the only place where we report meV/atom instead of meV. In general, meV/atom is not an ideal metric for molecular systems - it stems from material structures which contain periodicity. Mentions of 'Fock' have now been removed, and only the term 'Hamiltonian' is used.
>
> [Question 1]: It will not be possible to finetune a model trained on def2-SVP to def2-TZVPD, because there is a different maximum angular degree: $l_{max}$ = 4 for $\nabla^2$DFT, while $l_{max}$ = 6  for OMol\_CSH\_58k. This means the embeddings in the backbone need to contain irreps up to l=6 to match the dimensions of the data. Architecturally speaking, it is possible to do the reverse, e.g., to finetune a model pretrained on a larger basis, if some data is discarded. Whether this is beneficial is something we cannot say without extensive experiments.

---

> > ### Author Response · Authors · 2025-11-25
> > **Addressing the last three questions**
> >
> > **[Question 2]**: Yes, it is possible either to use a direct-force output head or to use conserving forces from an energy output head. In this work we use energies for the prediction task because they can be computed directly with the Hamiltonian matrix. Finetuning on conserving forces computed through a second backpropagation step from the energy head is a part of our future work.
> >
> > **[Question 3]**: Since submission, HELM has since been made capable of treating open shell systems. Along with modifications to the data processing pipeline, the necessary architectural modifications were (1) charge and spin embeddings in the backbone (included in the node initialization), and (2) the use of spin-resolved learnable weights in the Hamiltonian head.
> >
> > **[Question 4]**: We have extracted the requested computational performance metrics for HELM in Hamiltonian inference mode (these include both the node and edge update blocks, and are thus the computationally-heaviest version of HELM). We do not currently have access to V100 or A100 GPUs, so we report metrics on H100s and H200s instead, which are available on the cluster we used for the results reported in the paper.
> >
> > > Metric 1: Training - Time per epoch: We report data for the $\nabla^2$DFT-2k train split using 64 molecules per batch (as used for the results in the paper). As in the paper, training occurs in a distributed environment on 64 GPUs, with gradient reduction managed by PyTorch's Distributed Data Parallel (DDP) framework using the NCCL backend. The timing measurements per batch thus include the effects of load imbalance (we do not use the edge-wise batching protocol mentioned in Appendix C.2 for $\nabla^2$DFT, as the molecules are more homogeneous in size) and communication overhead. In all cases, we record times for 50 epochs and report the median and 90\% confidence intervals (5th percentile, 95th percentile) excluding the first 10. For this dataset and batch size, the peak GPU memory usage per epoch is 6.79 GiB.
> >
> > | | **H100 (s)** | **H200 (s)** |
> > | :--- | :---: | :---: |
> > | Time per Epoch (s) | 55.8480 [52.6620, 59.7237] | 42.1806 [40.4673, 44.8934] |
> >
> > > Metric 2: Training - GPU memory usage with $l_{max}$: We cannot vary $l_{max}$, as when training on orbital interactions from the Hamiltonian matrix, $l_{max}$ is **not a hyperparameter**. This is because the orbital interaction data is transformed into a direct sum of rank-0 to rank-$N$ tensors. $l_{max}$ must be set to $N$, otherwise the dimension of the model outputs will be incorrect. We note that all the evaluations in this paper were done on a single H100 GPU with 80GB memory, even for $l_{max} = 6$ and the largest molecules in the OMol\_CSH\_58k dataset.
> >
> > > Metric 3: Inference time per molecule: As requested, we report the Hamiltonian inference time per molecule for molecules of various sizes in the $\nabla^2$DFT 2k split. The process of `inference' for Hamiltonian matrices requires (1) passing the atomic structure graph through HELM's background to collect the embeddings, (2) passing these embeddings through the Hamiltonian output head to convert them into a direct sum of irreps corresponding to the padded targets, (3) performing a basis transformation on these targets to convert them to the uncoupled space of orbital interactions, and (4) unpadding these targets and using them to reconstruct the predicted matrix block-by-block. We report timing details for each of these four stages, along with the total time. The time taken depends on both the number of atoms and the number of inter-atomic interactions (graph edges) in the molecule, so there is variability in the data, which we indicate with the 90\% confidence intervals  ($\pm$) .
> >
> > | | **Molecule Size** | **Backbone** | **Head** | **Basis Transform** | **Matrix Recon.** | **Total** |
> > | :--- | :---: | :---: | :---: | :---: | :---: | :---: |
> > | ***H100*** | | | | | | |
> > | | 20--30 | 0.0609$\pm$0.0109 | 0.0146$\pm$0.0006 | 0.0272$\pm$0.0012 | 0.0462$\pm$0.0108 | 0.1565$\pm$0.0147 |
> > | | 30--40 | 0.0622$\pm$0.0140 | 0.0147$\pm$0.0012 | 0.0274$\pm$0.0011 | 0.0651$\pm$0.0138 | 0.1780$\pm$0.0216 |
> > | | 40--50 | 0.0596$\pm$0.0128 | 0.0148$\pm$0.0009 | 0.0282$\pm$0.0023 | 0.0890$\pm$0.0220 | 0.2002$\pm$0.0290 |
> > | | 50--60 | 0.0601$\pm$0.0132 | 0.0149$\pm$0.0020 | 0.0289$\pm$0.0040 | 0.1220$\pm$0.0236 | 0.2377$\pm$0.0254 |
> > | ***H200*** | | | | | | |
> > | | 20--30 | 0.0444$\pm$0.0080 | 0.0085$\pm$0.0006 | 0.0174$\pm$0.0005 | 0.0353$\pm$0.0076 | 0.1146$\pm$0.0106 |
> > | | 30--40 | 0.0463$\pm$0.0113 | 0.0087$\pm$0.0008 | 0.0176$\pm$0.0008 | 0.0494$\pm$0.0119 | 0.1318$\pm$0.0188 |
> > | | 40--50 | 0.0441$\pm$0.0103 | 0.0086$\pm$0.0009 | 0.0181$\pm$0.0014 | 0.0680$\pm$0.0208 | 0.1493$\pm$0.0282 |
> > | | 50--60 | 0.0444$\pm$0.0033 | 0.0087$\pm$0.0005 | 0.0184$\pm$0.0005 | 0.0941$\pm$0.0234 | 0.1765$\pm$0.0223 |

---

### Official Review · Reviewer_7bnb · 2025-10-31

**Soundness:** 3
**Presentation:** 3
**Contribution:** 3
**Rating:** 8
**Confidence:** 4

**Summary:**

This paper introduces HELM, a scalable equivariant graph neural network that learns from Hamiltonian matrices and proposes “Hamiltonian pretraining” as a route to enhance molecular property prediction. The idea is well-motivated, technically sound, and experimentally verified on large and diverse datasets. The proposed OMol CSH 58k dataset is likely to have significant value for the community.

**Strengths:**

1. Strong and well-motivated idea — leveraging electronic-structure information that is usually discarded in MLIP training is both physically meaningful and practically impactful.
2. Technical soundness — the HELM architecture is carefully designed with rotational equivariance and SO(2) convolutions for scalability to large basis sets and diverse elements.
3. Dataset contribution — the new OMol CSH 58k dataset fills a gap between small-molecule Hamiltonian data and modern large-scale MLIP datasets.

**Weaknesses:**

1. Conceptual novelty is moderate: the idea of predicting or using Hamiltonian matrices has been explored in literature; this paper mainly provides a unified and scalable realization.
2. Performance improvements are modest: while pretraining improves energy prediction in low-data regimes, the gain is typically 1.5–2×, and the benefit diminishes for larger datasets.
3. No demonstration on forces or MD trajectories: the method is not yet shown to improve dynamical stability or energy–force consistency, which are critical for MLIP applications.
4. Dataset construction: The details about the settings of how the data points were computed are not clear to me. The authors claimed that the dataset expands over many elements in the periodic table. However, Gaussian basis sets calculations are unknown to fail for many heavy elements. So, how did you handle these issues, to be specific, how to choose basis or effective core potentials for those molecules?

**Questions:**

The idea of pretraining on Hamiltonian matrices is certainly reasonable and physically motivated. However, this approach requires maintaining detailed edge-level information in the molecular graph, which could substantially increase memory usage. In particular, traditional MLIPs often rely on relatively low-order spherical harmonics for edge features, whereas predicting Hamiltonian elements likely demands much higher angular degrees to capture orbital interactions. This would significantly raise both computational and memory costs.
Could the authors comment on this trade-off? Specifically, how does the computational and memory footprint of HELM compare to that of conventional MLIPs trained only on energies and forces? Moreover, under the same computational budget, does HELM still outperform existing energy-prediction models?

---

> ### Author Response · Authors · 2025-11-25
>
> Thank you for your evaluation of our paper! We include some discussion on the 'weakness' points and question below:
>
> Weakness 1: Along with a scalable realization, we introduce careful implementation of node-parity constraint blocks and symmetry-preserving data scaling methods. Both of these are necessary to treat heavy elements (we have added ablations in our response to Reviewer atd4 to illustrate that these make up a bulk of the improvements leading to SOTA Hamiltonian prediction).
>
> Weakness 3: Our primary contributions are (1) a Hamiltonian model which significantly pushes SOTA in Hamiltonian prediction, (2) a curated Hamiltonian dataset of unprecedented scale, and (3) introduction of Hamiltonian-pretraining as a general approach for electronically-informed molecular property prediction. The natural next steps for us are to finetune with conserving-forces to additionally tackle MLIP applications, but it was beyond the scope of the current work!
>
> Weakness 4: Please see the Open Molecules 2025 paper [1] for details on the dataset generation! OMol\_CSH\_58k is a curated subset of the Hamiltonian matrices produced when generating OMol25. The def2-TZVPD basis set was employed throughout, with the def2 ECP being applied for heavy elements.
>
> [1] Daniel S. Levine, et. al., The open molecules 2025 (omol25) dataset, evaluations, and models, 2025. URL https://arxiv.org/abs/2505.08762.
>
> Question:
>
> There are two components to the question, which together generally make Hamiltonian prediction models computationally heavier than energy prediction models:
>
> (1) Edge-level information: This point may already be clear, but we clarify just in case there is some ambiguity: The models in Table 4 all contain the same number of parameters, and thus already have the same time per forward pass. This is because we omit the edge embedding update blocks after the Hamiltonian pretraining phase, so the edge-level information is maintained only indirectly within the node blocks. The exact number of parameters in the backbone and output head are given in Table 5 of Appendix C.1. So the necessity of incorporating edge-level embeddings for the Hamiltonian pretraining phase does not, in itself, raise the computational cost of subsequent use in energy prediction.
>
> (2) Spherical harmonic degree: We require $l_{max}$ of 4 ($\nabla^2$DFT) and 6 (OMol\_CSH\_58k) to match the dimensions of the orbital interactions in these datasets. These are, indeed, larger than what is used by many MLIPs.
>
> The use of eSCN convolutions [2] (which have a reduced computational complexity with $\mathcal{O}(l_{max}^3)$, rather than the $\mathcal{O}(l_{max}^6)$ of tensor products) is what makes these high $l_{max}$ currently feasible. Models such as UMA [3] similarly use $l_{max}$ of 4/6 (medium/large models, although note that they use lower $m_{max}$). As the rank-0 energies and rank-1 forces approximated by MLIPs are a property of the underlying electronic interactions, which contain data of up to rank-6 for this basis, we think that using optimized eSCN kernels to learn from large volumes of high-rank Hamiltonian data is a promising direction to learn molecular properties to higher generalizability. We do not yet have comparisons to report with other MLIPs, but (1) further reducing the computational budget of HELM and (2) training/finetuning on Hamiltonian matrices/force/energy data at scale is something we are actively working on.
>
> To provide an idea of the computational cost of HELM, in our response to reviewer RqiD we have included some performance metrics taken in Hamiltonian prediction mode, where both edge-level updates and the Hamiltonian head are included in the runtimes.
>
> [2] Saro Passaro and C. Lawrence Zitnick. Reducing SO(3) convolutions to SO(2) for efficient equivariant GNNs, 2023. URL https://arxiv.org/abs/2302.03655.
>
> [3] Brandon M. Wood, et. al, UMA: A family of universal models for atoms, 2025. URL https://arxiv.org/abs/2506.23971.

---

### Official Review · Reviewer_atd4 · 2025-11-01

**Soundness:** 2
**Presentation:** 3
**Contribution:** 3
**Rating:** 4
**Confidence:** 3

**Summary:**

This paper presents HELM, a novel approach for predicting molecular potential energies with the help of large amounts of electronic Hamiltonian matrix data, and a curated Hamiltonian matrix dataset OMol_CSH_58k. HELM adopts a novel backbone model architecture, first pre-trains model to predict Hamiltonian matrix then trains model to predict energy. Experiments show HELM achieves better accuracy than baselines in Hamiltonian prediction and good energy prediction accuracy.

**Strengths:**

- Propose a novel approach for molecule energy prediction. The idea of first pre-training on Hamiltonian prediction task then training on energy prediction task is quite useful and generalizable to multiple different models.
- Experiment results demonstrate good performance of the proposed HELM approach.
- The writing of this paper is clear and well-organized.

**Weaknesses:**

- Need clarification on results comparison in Table 2: are HELM and all baselines in Table 2 pre-trained on the OMol_CSH_58k Hamiltonians to ensure fair comparison? If so, authors are encouraged to highlight what is the advantages of the architecture design in HELM over baselines that leads to better Hamiltonian prediction accuracy.
- In Table 2, there is no baseline comparison in energy prediction so there is no idea how good the performance of HELM is. Authors are suggested to add baseline performance. If baseline methods are using the predicted Hamiltonian matrices to compute energy, compare with the energy prediction accuracy of HELM under the same setting.

**Questions:**

No additional questions.

---

> ### Author Response · Authors · 2025-11-25
> **Ablations to highlight the advantages of HELM for Hamiltonian prediction**
>
> Thank you for reviewing our paper! We have included responses to each question (weaknesses) below, to clarify data and provide additional ablations.
>
> 1. The comparisons in rows 1-5 of Table 2 (which include both HELM and other models) are on Hamiltonian matrix benchmarks. There is no pretraining at this stage, because the evaluation targets are the Hamiltonian matrices of the corresponding datasets (MD17 and $\nabla^2$DFT). HELM is state-of-the-art in Hamiltonian prediction.
>
> We acknowledge that, while we described our optimizations in the paper, we missed including ablations to highlight their advantages. The major architectural differences between HELM and the other models shown in Table 2 are (1) coupled node/edge updates, (2) the use of eSCN convolutions, (3) parity-restrictions of the node outputs (node parity expansion layer), and (4) symmetry-preserving scaling/unscaling modules. All of these are not present in the other models shown in Table 2, which are based on either symmetrically unconstrained or SO(3) equivariant networks.
>
> The coupled node/edge updates in (1) are novel to HELM, and were designed to enable transferability to downstream energy prediction tasks. They are not intended specifically to improve Hamiltonian prediction accuracy. Next, the advantage of eSCN convolutions in isolation is not novel to HELM, and has been studied in Ref [1], where they were able to get lower matrix MAE errors on QH9 and a custom dataset. So there remains differences (3) and (4) which should be ablated to highlight their advantages. The effect of (3) can be isolated by simply omitting the node parity expansion step and predicting all the diagonal submatrix elements of the Hamiltonian independently. The effect of (4) can be isolated by omitting the scaling procedure and using the raw matrix elements for training.
>
> We report the results of these ablations below using the $\nabla^2$DFT 2k train/test splits. All of these were done with a fixed number of epochs (500) and a cosine scheduler for fair comparison between different settings. Note that this is not long enough to fully converge the training (which was done using a ReduceLRonPlateau scheduler for the results in Table 2 of the paper, and converged at $\sim$1200 epochs for the 2k split).
>
> | $l=0$ Scaling | Node-Parity Expansion | $H^{err}$ - train ($\times 10^{-6} E_h$) | $E^{err}$ - train (meV) | $H^{err}$ - test ($\times 10^{-6} E_h$) | $E^{err}$ - test (meV) |
> |---------------|-----------------------|-------------------|-------------------|------------------|------------------|
> | ×             | ×                     | 201.21            | 169,026.08        | 324.35           | 329,923.53       |
> | ×             | ✓                     | 95.73             | 1,163.54          | 101.73           | 1,530.83         |
> | ✓             | ×                     | 115.35            | 837.07            | 120.18           | 973.47           |
> | ✓             | ✓                     | 92.00             | 540.19            | 96.27            | 706.39           |
>
> The node parity expansion block only impacts the error within diagonal blocks of the Hamiltonian. These diagonal blocks contain the largest-magnitude elements (resulting from nearsightedness of orbital interactions), and contribute disproportionately towards the eigenvalue error and total energy prediction. Therefore, improving the loss over diagonal orbital blocks has a significant impact on the total energy computed from the matrices, bringing the error down from 329,923 meV to 1,530 meV. The symmetry-preserving scaling procedure serves to reduce the numerical range of the data from the order of $\sim$100 to the order of $\sim$1 (see Fig. 7 in Section B.3), thus making it easier to learn with a properly-normalized model. This brings the energy error down further to 706 meV.
>
> We mention in the paper that HELM is designed to train on large Hamiltonian datasets. The properties of the data which result in these differences (e.g., large magnitude diagonal elements) are drastically worsened when heavy elements are included (see Fig. 7), and these optimizations become essential. Specifically, the results in the table above are on $\nabla^2$DFT, the second largest Hamiltonian dataset, which contains only 1 `heavier' element (Br, Z=35). Meanwhile, OMol\_CSH\_58k contains many of the heaviest elements (Z up to 83), which are all well-represented with $\sim$ 1,000 occurrences each.
>
> [1]  Li Y, Xia Z, Huang L, et al. Enhancing the scalability and applicability of kohn-sham hamiltonians for molecular systems. arXiv preprint arXiv:2502.19227, 2025.
>
> 2. We clarify that in the last row of Table 2, going from the Hamiltonian matrix to the energy does not involve any learnable parameters - it is possible to compute energy directly from the matrix (see Appendix A.2), but this is not a common way to predict energies. We did these calculations to compare them later with energy finetuning (Table 3).

---

> > ### Comment · Reviewer_atd4 · 2025-11-25
> > **Follow-up Response**
> >
> > I appreciate authors' efforts in rebuttal. My concerns have been addressed so I increase my rate.

---

### Author Response · Authors · 2025-11-25
**A summary of our (major) comments:**

1. Ablations of model components for Hamiltonian prediction: As requested by Reviewers atd4 and RqiD, we included additional data to show that improvements in Hamiltonian prediction stem primarily from the node-parity expansion layers and symmetry-constrained data scaling integrated into HELM. This data was also used to address Reviewer QD9Z's question about why we do not directly observe the SAD effect from Ref [1] in our model outputs.

2. Computational performance metrics: We report wall-clock time per training epoch, and inference for HELM on Hamiltonian matrices, as requested by Reviewer RqiD (on H100 and H200 GPUs rather than the two they requested, due to available resources). We clarified that we cannot freely vary $l_{max}$ during training as it is not a hyperparameter in our models, and is fixed to the dimensions of the orbital interaction data.

3. Additional clarifications made: We clarified with Reviewer RqiD that we used the functionals which match every dataset to compute total energy values, and clarified for Reviewer QD9Z that the trends in Fig. 3(c) and Table 4 are indeed consistent with each other, as they expected. We suspect the latter might have been confusing because Tables 3 and 4 are the `transpose' of each other.

[1]  Li Y, Xia Z, Huang L, et al. Enhancing the scalability and applicability of kohn-sham hamiltonians for molecular systems. arXiv preprint arXiv:2502.19227, 2025.

---

### Author Response · Authors · 2025-11-29
**Summary comment**

We summarize (1) results of our single round of discussion with reviewers, (2) clarifications made and (3) new data we provided in our responses, referring only to content available below.

**Notable quote:** "I do not find any obvious weaknesses in this paper." - Reviewer QD9Z (confidence score 5)

**Results of discussion:** Two out of four reviewers (atd4, QD9Z) have responded to our reply. Both of them replied that their questions have been addressed, so they intend to increase their score.

**Clarifications we made in our responses to reviewers:**

- Clarified with Reviewer atd4 (confidence score 3) that (1) the benchmarks in Table 2 are for Hamiltonian prediction and thus do not use Hamiltonian pre-training, and (2) previous MLIPs did not predict energies by computing them from the Hamiltonian matrix.

- Clarified with reviewer QD9Z (confidence score 5) that in Table 4, the order of results is "finetuned < pretrained-frozen", consistent with Fig 3, and not "pretrained-frozen < finetuned" as they read and thus questioned. We believe this misunderstanding occurred because the headings of Tables 3 and 4 are transposed for better visual layout.

- Clarified with Reviewer RqiD (confidence score 5) that we used the correct functional for each dataset when computing the total energy (they misunderstood that we used wb97m-v for all three benchmarks, MD17, $\nabla^2$DFT, and OMol_CSH_58k, because we made a reference to wb97m-v within a more general appendix section).

- Clarified with Reviewer 7bnb (confidence score 4) that the models we finetuned have the same number of parameters as the ones used for direct energy training, and thus the same computational cost. Note that this was a part of their more general question on computational comparisons between MLIPs and Hamiltonian models, to which our complete answer is below.

**New data collected:**

- Two out of four reviewers asked if we could ablate specific model components of HELM to isolate why it achieves SOTA performance on Hamiltonian prediction benchmarks. We collected ablation studies to demonstrate that beyond the addition of eSCN convolutions, our node parity expansion block and symmetry-constrained scaling procedure result in (1) state of the art matrix element errors and (2) significantly low errors in the derived total energy. We explain why this is the case in our reply to Reviewer atd4.

- We report timing measurements for $\nabla^2$DFT training and inference on two different GPUs for Reviewer RqiD

We have satisfied all reviewer questions (in both the questions and weakness sections), with the exception of Reviewer RqiD's suggestion for ablation studies with dropout, layer-freezing and weight decay (due to time constraints for re-training at this level).

We will be available for further discussion.

---

### Meta-Review · Area_Chair_7egN · 2026-01-05

**Summary:**

HELM ('Hamiltonian-trained Electronic-structure Learning for Molecules') is reported to achieve state-of-the-art Hamiltonian as well as downstream energy prediction performances by leveraging scalable eSCN. In addition, they also provided a new and larger Hamiltonian matrix dataset, 'OMol_CSH_58k'.

**Reviewer Concerns:**

Based on available discussions, the authors have to clarified most of the reviewers' concerns, including:

1. Limited novelty considering using existing eSCN: the authors clarified that leveraging structure properties from Hamiltonian is also critical for achieving state-of-the-art performances.

2. Additional data as well as experimental results, including new ablation studies, were provided during discussion to address the concerns on benchmarking experiments.

Reviewer 7bnb requested the force prediction results but the authors stated that "it was beyond the scope of the current work!"

Review RqiD asked for more rigorous analysis exploring regularization strategies but the authors did not address this but considered as future work.

The authors may want to consider including these additional experimental results in the final version to further improve the paper.

**Reviewer Scores:**

Based on the available discussions and the summary from the authors, the reviewers will either maintain their scores or increase from 4 to 6 (reviewers atd4, QD9Z). Therefore, after the review-author discussions, the reviewers have reached consensus that this is a good paper to be presented in ICLR.

---

### Decision · Program_Chairs · 2026-01-26

Accept (Poster)